# From Growing to Looping: A Unified View of Iterative Computation in LLMs

Ferdinand Kapl [* 1 2]   Emmanouil Angelis [* 1 2 3]   Kaitlin Maile [† 4]   Johannes von Oswald [† 4]   Stefan Bauer [† 1 2 3]

## Abstract

Looping, reusing a block of layers across depth, and depth growing, training shallow-to-deep models by duplicating middle layers, have both been linked to stronger reasoning, but their relationship remains unclear. We provide a mechanistic unification: looped and depth-grown models exhibit convergent depth-wise signatures, including increased reliance on late layers and recurring patterns aligned with the looped or grown block. These shared signatures support the view that their gains stem from a common form of iterative computation. Building on this connection, we show that the two techniques are adaptable and composable: applying inference-time looping to the middle blocks of a depth-grown model improves accuracy on some reasoning primitives by up to $2\times$, despite the model never being trained to loop. Both approaches also adapt better than the baseline when given more in-context examples or additional supervised fine-tuning data. Additionally, depth-grown models achieve the largest reasoning gains when using higher-quality, math-heavy cooldown mixtures, which can be further boosted by adapting a middle block to loop. Overall, our results position depth growth and looping as complementary, practical methods for inducing and scaling iterative computation to improve reasoning.

## 1. Introduction

The dominant paradigm for improving the performance of Large Language Models (LLMs) has been to scale both parameters and data simultaneously (Kaplan et al., 2020; Hoff-

mann et al., 2022). However, reasoning capabilities, which are often framed as the ability to perform multi-step logical deductions, do not always scale linearly with parameter count alone (Wei et al., 2022a; Xu et al., 2025). A popular way to increase reasoning performance at inference time is to generate longer textual traces via chain-of-thought (Wei et al., 2022b), but this scales compute through *tokens* and does not directly encourage *internal* computation. A complementary direction is to build iteration into the model's forward pass: repeatedly applying transformations in latent space so representations can be refined over multiple steps before producing the final prediction (Saunshi et al., 2025; Zhu et al., 2025a;b; McLeish et al., 2025; Geiping et al., 2025; Koishekenov et al., 2025; Bae et al., 2025b).

A prominent approach in this direction is *looped* models (Universal Transformers), motivated by the goal of decoupling model size from computational depth. By tying weights across layers and executing the same block recurrently, looped models can perform deeper computations without increasing the number of unique parameters (Dehghani et al., 2019; Lan et al., 2020; Dabre & Fujita, 2019). While early investigations focused mostly on parameter efficiency, recent findings highlight their potential for reasoning. Previous work (Saunshi et al., 2025; Geiping et al., 2025; Zhu et al., 2025b) observed that looped models can scale latent computations to increase reasoning performance. Similarly, approaches like Wang et al. (2025); Jolicoeur-Martineau (2025) utilize recursive architectures to maximize the reasoning capacity of notably small networks.

*Growing* models, an alternative paradigm for training, was originally motivated by increased training efficiency through parameter growth (Gong et al., 2019; Wang et al., 2023; Reddi et al., 2023). By initializing a shallow model and progressively adding layers during training, the total training FLOPs required to reach a target depth can be significantly reduced. Additionally, recent work discovered an intriguing inductive bias. Saunshi et al. (2024) and Kapl et al. (2025) demonstrated that models trained via a particular type of growing, i.e., via duplication of blocks in the middle of the transformer architecture, outperform equally sized baselines trained from scratch on reasoning tasks, even when controlling for final architecture and training dataset composition.

**The Connection: A Unified View?** Depth growing creates

---

[*]Equal contribution [†] Provided equal in-depth feedback and guidance. [1]Technical University of Munich [2]Helmholtz AI, Munich [3]Munich Center for Machine Learning (MCML) [4]Google, Paradigms of Intelligence Team. Correspondence to: Ferdinand Kapl <ferdinand.kapl@tum.de>, Emmanouil Angelis <emmanouil.angelis@tum.de>.

*Proceedings of the 43rd International Conference on Machine Learning*, Seoul, South Korea. PMLR 306, 2026. Copyright 2026 by the author(s).

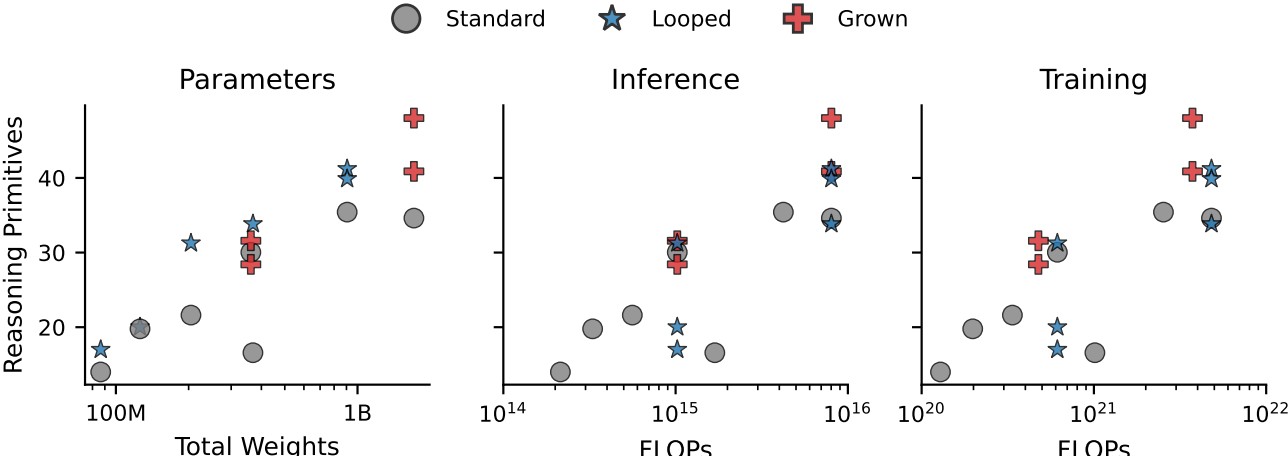

*Figure 1.* **Trade-offs for looped and depth-grown models.** Each point corresponds to a model in Table 1 (up to 1.7B parameters), plotted by average *Reasoning Primitives* accuracy versus unique parameters (left), inference FLOPs (middle), and training FLOPs (right). Looped and depth-grown models improve accuracy in reasoning primitives over *standard* baselines, suggesting a shared inductive bias toward better reasoning. Looped models improve reasoning under fixed parameter budgets and can be competitive under fixed inference budgets, while depth-grown models reach similar or better reasoning with less training compute. Figure 10 shows additional benchmark categories.

*implicit* repetition through initialization (duplicating layers), while looping enforces *explicit* repetition through weight tying. This raises a fundamental question: Do the observed reasoning gains in depth-grown and looped models arise from the same underlying mechanism? Beyond the superficial resemblance implied by parameter-level block similarity of grown models (Saunshi et al., 2024; Kapl et al., 2025), the relationship between the *grown* structure and *looped* execution remains underexplored.

**Contributions.** In this work, we provide a mechanistic unification of looping and depth growing, and show how to compose them for scalable reasoning.

- **Empirical trade-offs.** We benchmark standard, looped, and depth-grown transformers at 360M and 1.7B across 22 tasks, and characterize the resulting trade-offs between unique parameters, inference FLOPs, and training FLOPs for reasoning performance. We find that looped models improve reasoning under fixed parameter budgets and can be competitive under fixed inference budgets, while depth-grown models reach similar or better reasoning with less training compute.

- **Unified mechanistic signatures.** Using depth-utilization diagnostics and residual-stream interventions, we show that looped and depth-grown models exhibit similar depth-wise computational signatures. Both approaches shift indispensable computation to later layers and induce periodic, block-aligned patterns in residual updates and sublayer contributions, consistent with iterative computation.

- **Intervention robustness and looping in the middle.** Through layer-swapping interventions, we find fully

looped models are more order-sensitive, while looping in the middle designs, empirically the best in our setting, with unique encoder–decoder layers recover robustness similar to depth-grown models, suggesting a practical design principle for tying recurrence to the middle of the network.

- **Adaptability.** Looped and depth-grown models adapt more efficiently than standard baselines under in-context learning and supervised fine-tuning. Furthermore, depth-grown models achieve the largest gains when exposed to high-quality, math-heavy data mixtures during cooldown.

- **Composability: Grow first, loop later.** Depth-grown models can be looped at inference time by repeating a middle block, yielding up to $2\times$ gains on reasoning primitives despite never being trained with weight tying. Further, retrofitting recurrence to `LIDAS` during cooldown, combined with a high-quality math-focused cooldown mixture, produces the strongest reasoning performance under matched data and inference FLOPs.

Overall, our results suggest that growing and looping, despite their different origins, are complementary methods for inducing and scaling iterative computation for reasoning.

## 2. Related Work

**Growing Neural Networks.** Growing reuses a smaller model to initialize a deeper one, reducing pre-training compute. Common approaches include copying existing layers to initialize newly added depth (Du et al., 2024; Kim et al., 2024; Gong et al., 2019; Reddi et al., 2023), learning a mapping (Wang et al., 2023), or masked structural growth

(Yao et al., 2024). Saunshi et al. (2024) introduce MIDAS, which partitions the network into blocks and duplicates the middle block rather than the end, preserving efficiency while improving reasoning at comparable perplexity. Kapl et al. (2025) analyze depth-grown models, including MIDAS, with depth diagnostics and block-level interventions. They argue that growth changes depth-wise computation and makes later layers more indispensable. Additionally, they propose LIDAS, which duplicates the exact layer-wise middle, yielding a more symmetric weight structure and stronger reasoning performance.

**Looped and recurrent models.** Depth-wise parameter sharing and recurrence have been used to trade unique parameters for computation, from Universal Transformers (Dehghani et al., 2019) and recurrent seq2seq (Dabre & Fujita, 2019) to ALBERT (Lan et al., 2020) and broader sharing analyses (Takase & Kiyono, 2023). Recent variants revisit recurrence with partial sharing and added capacity, such as using low-rank adapters (Bae et al., 2025a) or Mixture-of-Experts (Csordás et al., 2024). Looped models are broadly used for their iterative computation and test-time scaling. They improve in-context learning-to-learn (Yang et al., 2024) and algorithmic length generalization (Fan et al., 2025) and enable latent reasoning (Saunshi et al., 2025; Zhu et al., 2025a). Additional works investigate looped pre-training at scale (Zhu et al., 2025b; Geiping et al., 2025), and motivate converting fixed-depth pre-trained models into recurrent ones (McLeish et al., 2025; Koishekenov et al., 2025). Related non-classical transformer settings include hierarchical multi-timescale computation (Wang et al., 2025) and compact recursive refinement architectures (Jolicoeur-Martineau, 2025).

# 3. The Inductive Bias of Looped and Depth-Grown Models

Training Transformer-based LLMs by *depth growing* (Saunshi et al., 2024; Kapl et al., 2025) and *looping* (Saunshi et al., 2025; Zhu et al., 2025b; McLeish et al., 2025; Geiping et al., 2025; Koishekenov et al., 2025) has been argued to improve reasoning performance. Depth-grown models are trained by progressively increasing depth via middle-layer duplication, yielding strong reasoning performance at reduced training compute. Looped models, in contrast, explicitly tie weights across depth and iteratively apply a small set of layers, trading *unique parameters* for additional sequential computation. Despite this connection, their practical trade-offs and the extent to which they share a common inductive bias toward reasoning have only been alluded to (Kapl et al., 2025; Saunshi et al., 2025) and remain unclear. The following sections introduce notation for looped and depth-grown models (§3.1) and compare their performance under parameter, inference, and training budgets (§3.2).

## 3.1. Notation of Looped and Depth-Grown Models

We fix a Transformer architecture class (width, heads, embedding size, tokenizer, etc.) and vary only depth and parameter sharing. Let $f_L = [\ell_0, \ldots, \ell_{L-1}]$ denote a standard $L$-layer Transformer with *untied* parameters across layers, excluding the embedding matrix and the final LM head for simplicity.

**Looped models.** A looped model reuses a sequence of consecutive layers (the *recurrent block*) by repeatedly applying this fixed set of layers. We denote by $\mathrm{Loop}\,(L{\times}k)$ a model with $L$ unique layers that is unrolled for $k$ repetitions, yielding an effective depth $L \cdot k$. For example, $\mathrm{Loop}(4{\times}6)$ repeats a 4-layer block 6 times to reach a final depth of 24 layers. We additionally consider designs that keep untied, i.e., unique, encoding–decoding blocks and loop only the middle recurrent block, written as $\mathrm{Loop}\,(e\text{-}L{\times}k\text{-}d)$. For instance, $\mathrm{Loop}\,(4\text{-}4{\times}4\text{-}4)$ has a 4-layer unique encoding block, a 4-layer recurrent block repeated 4 times, and a 4-layer unique decoding block (12 unique layers, 24 effective depth).

**Depth-grown models.** Depth growing starts from a shallow network and repeatedly applies a growth operator according to a fixed growing schedule that inserts new layers at mid-depth by duplicating existing layers. Concretely, we consider MIDAS (Saunshi et al., 2024), which duplicates the middle *block* (here with block size $B = 4$), and LIDAS (Kapl et al., 2025), which duplicates the exact layer-wise middle, which has been empirically shown to be a better growing operator. We refer to Appendix A for more details on growing. After the last growing operation, both methods reach the same final depth as the baseline: training models by growing in depth only changes the training procedure, not the final weight structure. Therefore, growing reduces training FLOPs versus the baseline because early stages have fewer layers. Note, that in contrast to looped models, all layers are always kept *untied* at training and inference time for *depth-grown* models.

## 3.2. Trade-offs of Looped and Depth-Grown Models

**Setup.** We compare standard baselines against (i) depth-grown MIDAS and LIDAS models reaching the same final depth, (ii) looped models with varying numbers of unique layers but always the same effective depth (*iso-inference*), and (iii) their *iso-param* standard counterparts (same unique-layer count, but no looping). All models are based on the SmolLM-v1 (Ben Allal et al., 2024) 360M and 1.7B model configurations trained on approximately 200B and 400B tokens, respectively; for more details, see also the training protocol from Kapl et al. (2025), which we follow. We note that tuning the baseline training hyperparameters for depth-grown or looped models may improve performance, but we leave this for future work.

*Table 1.* **Performance comparison of standard transformer baselines, looped models, and two depth-grown models MIDAS (Saunshi et al., 2024) and LIDAS (Kapl et al., 2025) at 360M and 1.7B base model sizes.** Looped models often outperform *iso-param* baselines and are competitive with *iso-inference* baselines, especially for reasoning-heavy task categories such as Open-book Q&A, Math Word Problems and Reasoning Primitives. Depth-grown models match the baselines across most task categories with roughly 80% of the pre-training compute, while outperforming them on reasoning. This suggests a shared inductive bias toward reasoning for looped and depth-grown models. Best performance per model size in bold and looped model rows in gray.

| | | Params / FLOPs | Standard cooldown | | | | | | |
| --- | --- | --- | --- | --- | --- | --- | --- | --- | --- |
| | | | Holdout Set (NLL ↓) | Open-book Q&A (F1 ↑) | Closed-book Q&A (F1 ↑) | Lambada (Acc ↑) | HellaSwag (Acc ↑) | Math Word Problems (Acc ↑) | Reasoning Primitives (Acc ↑) |
| **360M** | Baseline | 32 / 32 | 2.18 | 22.89 | 14.50 | 43.35 | 39.97 | 3.69 | 30.04 |
| | MIDAS | 32 / 32 | 2.18 | 24.57 | 13.75 | 43.26 | 40.36 | **4.39** | 28.42 |
| | LIDAS | 32 / 32 | **2.16** | **26.63** | **14.57** | **44.03** | **40.58** | 4.36 | **31.58** |
| | Standard | 4 / 4 | 2.67 | 7.88 | 6.54 | 26.20 | 29.68 | 2.02 | 14.00 |
| | Loop (4×8) | 4 / 32 | 2.50 | 14.79 | 8.97 | 31.69 | 32.42 | 2.37 | 17.00 |
| | Standard | 8 / 8 | 2.47 | 13.32 | 8.65 | 32.37 | 32.27 | 2.37 | 19.78 |
| | Loop (8×4) | 8 / 32 | 2.38 | 16.20 | 11.12 | 34.89 | 34.81 | 1.81 | 20.02 |
| | Standard | 16 / 16 | 2.31 | 17.38 | 11.24 | 37.94 | 35.72 | 2.90 | 21.62 |
| | Loop (16×2) | 16 / 32 | 2.27 | 20.74 | 11.48 | 38.02 | 37.25 | 2.45 | 31.26 |
| **1.7B** | Baseline | 24 / 24 | **1.96** | 29.57 | 18.60 | 50.05 | 46.28 | 13.61 | 34.62 |
| | MIDAS | 24 / 24 | 1.97 | 28.80 | 18.43 | 50.81 | 46.19 | 15.91 | 40.88 |
| | LIDAS | 24 / 24 | **1.96** | **29.84** | **19.07** | **51.41** | **46.32** | **18.18** | **48.02** |
| | Standard | 4 / 4 | 2.29 | 14.31 | 10.41 | 34.02 | 35.78 | 2.19 | 16.58 |
| | Loop (4×6) | 4 / 24 | 2.12 | 23.68 | 15.10 | 42.40 | 41.35 | 3.35 | 33.84 |
| | Standard | 12 / 12 | 2.01 | 25.33 | 17.30 | 46.36 | 43.69 | 5.42 | 35.42 |
| | Loop (12×2) | 12 / 24 | 2.07 | 25.56 | 15.31 | 44.71 | 42.44 | 3.32 | 41.22 |
| | Loop (4-4×4-4) | 12 / 24 | 2.05 | 27.32 | 17.00 | 47.51 | 43.52 | 6.91 | 39.88 |

**Benchmarks.** We report negative log-likelihood (NLL) on a held-out SmolLM-Corpus validation set, and follow the aggregated knowledge, language, and reasoning suite of Saunshi et al. (2024); Kapl et al. (2025), overall spanning **22 benchmarks**:

- **Closed-book Q&A:** Knowledge benchmarks that test memorization without context (TriviaQA, TyDiQA-NoContext, NaturalQuestions, WebQuestions) evaluated zero-shot.

- **Open-book Q&A:** Reading comprehension benchmarks that test extracting information from given context (TyDiQA-GoldP, SQuADv2, DROP, QuAC, CoQA) evaluated zero-shot.

- **Text Completion/Language Modeling:** Lambada (Paperno et al., 2016) and HellaSwag (Zellers et al., 2019) evaluated zero-shot.

- **Math Word Problems:** Simple math benchmarks (SVAMP (Patel et al., 2021), ASDiv (Miao et al., 2020), MAWPS (Koncel-Kedziorski et al., 2016)) evaluated five-shot.

- **Reasoning Primitives:** Reasoning benchmarks from Saunshi et al. (2024) evaluated five-shot. These include copying words tasks, and inferring the final value of a variable after a sequence of several assignments. An

example would be: $a = 3, b = 8, c = a, d = b, c = ?$. For more details see Appendix D.1.

All evaluations use the language model evaluation harness (Gao et al., 2024).

**Results and trade-offs.** Table 1 reports aggregated performance, and Figure 1 summarizes the resulting trade-offs between (i) unique parameters, (ii) inference FLOPs, and (iii) training FLOPs for reasoning performance. Figure 10 provides the same trade-off view for additional benchmark categories across knowledge, language, and reasoning metrics. We make the following three observations.

First, under an *iso-param* comparison (same number of unique parameters), looping is consistently beneficial: looped models outperform equally sized standard models across most task categories, with the largest gains on reasoning-heavy task categories (Open-book Q&A, Math Word Problems, and Reasoning Primitives).

Second, under an *iso-inference* comparison (same effective depth), looped models typically underperform the full baseline on knowledge and general language modeling, reflecting the previously observed relationship of knowledge capacity and number of unique parameters (Zhu et al., 2025a;b). However, when retaining enough unique pa-

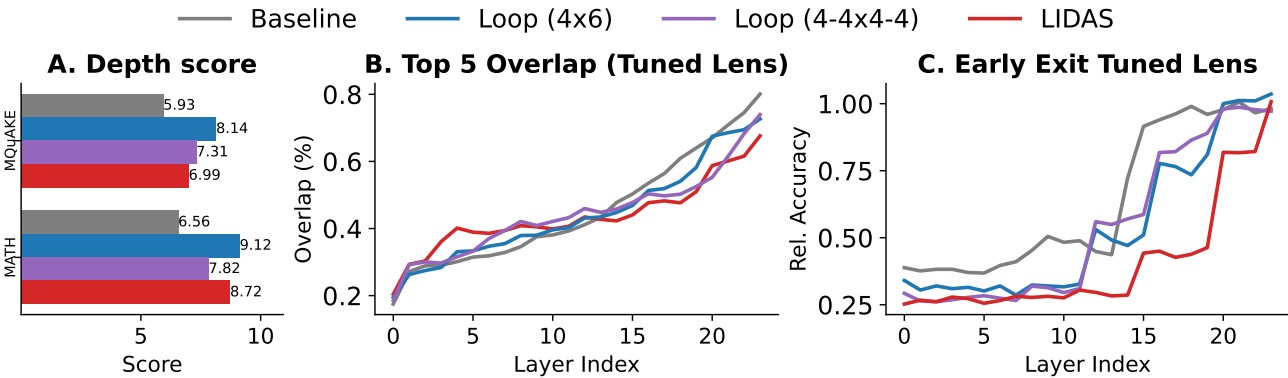

*Figure 2.* **Looped and depth-grown models use later layers more.** We compare Baseline, `LIDAS`, Loop (4×6) and Loop (4-4×4-4) on (A) depth score, (B) top-5 vocabulary overlap on GSM8K and (C) Tuned Lens early-exit normalized accuracy on the *Variable Assignment Math* reasoning primitive. All three diagnostics imply higher usage of later layers for the grown `LIDAS` and looped models.

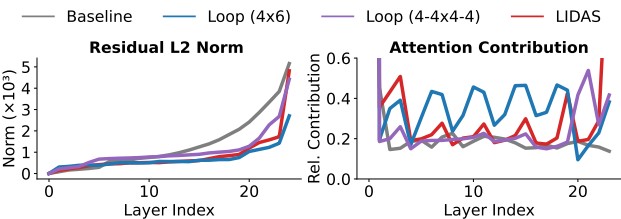

*Figure 3.* **Looped and depth-grown models exhibit similar (sub)layer usage.** `LIDAS`, Loop (4×6) and Loop (4-4×4-4) share a slower residual norm growth than the baseline and exhibit periodic attention-sublayer contributions (ratio of norms for attention sublayer output over residual) with a 4-layer cycle, matching the block size of `LIDAS` and the size of the recurrent block.

rameters, roughly 50% of the baseline, e.g. Loop (16×2) or Loop (12×2) for the 360M and 1.7B model, respectively, looped models exceed baseline accuracy on reasoning primitives. Across the looped variants considered here, Loop (4-4×4-4) provides the best overall performance.

Third, *depth-grown* models improve reasoning while preserving broad capabilities, and they do so at lower *training* compute (Kapl et al., 2025). In particular, `LIDAS` yields the strongest and most consistent reasoning gains while remaining superior or competitive on NLL, knowledge and language benchmarks (Table 1).

In summary, growing shifts the reasoning Pareto frontier toward lower training FLOPs in Figure 1 (right), while looped models are competitive at fixed inference budgets (middle) and offer a complementary trade-off when unique parameters are constrained (left).

## 4. The relationship of Looping and Growing

After empirically establishing that looped and depth-grown models share an inductive bias for better reasoning performance, we next investigate whether this co-occurs with

shared mechanistic traits.

In this section, we focus on the following 1.7B *iso-inference* variants: Baseline, the depth-grown `LIDAS` model (block size $B = 4$), and two looped models with matched effective depth but different numbers of parameters, Loop (4×6) and Loop (4-4×4-4). We investigate whether looping reproduces the depth-wise computational signatures previously attributed to depth growth (Kapl et al., 2025), increased usage of later layers, periodic (sub)layer patterns aligned with the block size, and robust computational blocks under layer-order interventions. Additional results and ablations are deferred to Appendix B.

### 4.1. Mechanistic Analysis

Kapl et al. (2025) recently observed that pre-training LLMs by growing in depth counteracts the *Curse of Depth* of standard pre-LayerNorm Transformers, where later layers contribute less to the final output distribution (Sun et al., 2025; Csordás et al., 2025). Here, we test whether explicit recurrence via looping yields depth-wise signatures that are similar.

**Does looping lead to higher depth usage?** To quantify whether later layers perform indispensable computation, instead of mostly small independent refinements, we use the depth-utilization diagnostics from Kapl et al. (2025): the depth score (Csordás et al., 2025), and both top-5 early-exit vocabulary overlap and early-exit accuracy via Tuned Lens (Belrose et al., 2023).

Across the three depth-utilization diagnostics in Figure 2, the two looped models (Loop (4×6), Loop (4-4×4-4)) closely track `LIDAS` and show substantially stronger late-layer reliance than the baseline. The grown and looped models have higher depth scores (Figure 2, left), indicating more indispensable computation in later layers. They also show lower top-5 overlap with the final prediction in later

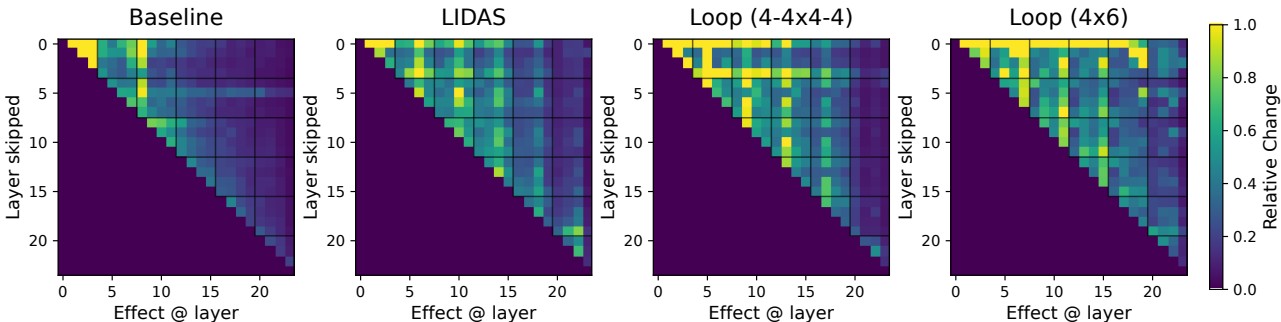

*Figure 4.* **Local effect of skipping a layer on downstream layer contributions for future tokens.** We intervene in the residual stream by removing the contribution of a layer from all subsequent layers separately (*local effect*) and measuring the relative change on the representations of all future tokens. `LIDAS` and the two looped models show a characteristic phenomenon. Every four layers, there is a layer that depends directly on most of the previous inputs (vertical pattern): removing the contribution of a previous layer results in relatively large changes in the representation of future tokens for this "aggregation" layer.

layers than the baseline (Figure 2, middle), suggesting that late layers continue to alter the model's outputs. Finally, for early-exit on *Variable Assignment Math*, the accuracy for the baseline plateaus earlier (Figure 2 right), whereas all other models continue improving their predictions until later layers.

**Residual stream and (sub)layer usage.** To connect these depth-utilization signals to internal dynamics, we first analyze residual stream norms and sublayer contributions. Then we intervene in the residual stream where we measure *local effects* by directly and separately removing the contribution of layers from future layers without propagating the changes. Comparing the Baseline, `LIDAS`, Loop (4×6) and Loop (4-4×4-4), we observe three recurring patterns. Both `LIDAS` and the two looped models show substantially slower growth of the residual stream norm than the baseline (Figure 3). Moreover, the contribution of the attention sublayer to the residual stream follows a clear 4-layer cycle, matching the block size of `LIDAS` and the recurrent block of the looped models. Finally, if we intervene in the residual stream by removing the output of previous layers from subsequent layers directly (*local effect*) without propagating the changes, local future-effect heatmaps exhibit an "aggregation"-like layer within each 4-layer segment that depends on a broad set of previous layers (Figure 4). Interestingly, the relative position of this layer within a block differs for all three models.

These shared patterns suggest that both training procedures encourage repeated, depth-periodic computation, and we include further ablations in Appendix B.3.

**Are looped models as robust as depth-grown models to layer interventions?** Kapl et al. (2025) observed robust permutable blocks in the middle of depth-grown networks. We investigate whether this robustness is likely due to the connection between looped and grown models, or whether

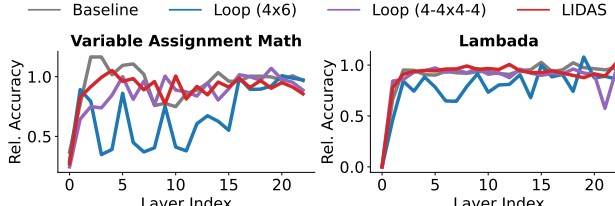

*Figure 5.* **Fully looped models are less robust to interventions.** Swapping a single layer degrades the performance of the fully looped model Loop (4×6) considerably on *Lambada* and *Variable Assignment Math* compared to the other models. Using unique encoder–decoder layers lets Loop (4-4×4-4) recover the robustness of the baseline and `LIDAS`.

it is more specific to the depth-growing mechanism.

In Figure 5, swapping even a single layer in the middle of the networks causes substantially larger performance degradation for the Loop (4×6) than for the baseline, `LIDAS`, or Loop (4-4×4-4). This also holds for larger interventions, e.g., swapping blocks of two consecutive layers, see Appendix B.1. Taken together, these intervention results suggest that, while looping and growing can yield similar depth-periodic (sub)layer usage, the fully looped model's computation is more order-sensitive, whereas depth growth can produce computational blocks in the middle that are comparatively more tolerant to local reorderings. Consistent with this, introducing unique encoder–decoder layers to the looped model design and looping a block in the middle of the network, here, Loop (4-4×4-4), recovers the robustness of the grown model. We therefore hypothesize that the depth-growing mechanism studied here, i.e., duplicating layers or blocks in the middle of the network, allows the model to flexibly learn unique encoder–decoder layers, with the middle of the network acting as a relaxed version of a fully looped or tied model.

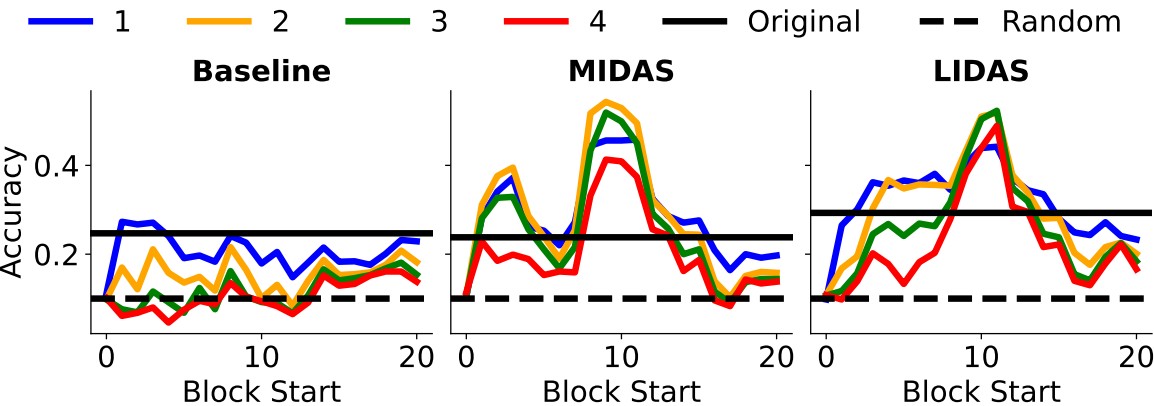

*Figure 6.* **Grown models benefit from looping at inference time.** Repeating a block of four layers in the middle of the network during inference increases the accuracy of grown models (MIDAS, LIDAS) on the *Copy Real Words* reasoning primitive up to $2\times$ compared to the original network. In contrast, the baseline rarely benefits from additional repetitions.

### 4.2. Inference Scaling

After confirming that looped and depth-grown models exhibit similar computational patterns internally, a natural question is: Can we loop depth-grown models at inference time to increase their reasoning performance? This is challenging in general, as even looped models trained with a fixed number of recursions often do not extrapolate to more recursions and instead degrade performance (Zhu et al., 2025b).

Since we consider grown models with block size $B = 4$, we focus on looping 4-layer blocks. Perhaps surprisingly, both grown models (MIDAS, LIDAS) can benefit greatly from repeating blocks of size 4 in the middle of the network to increase performance on a reasoning primitive up to $2\times$ (Figure 6), without ever being trained to loop. This is consistent with other reasoning primitives (Appendix C.3), where in general the baseline does not benefit at all or a lot less than the grown models from additional inference compute. Zooming in on the number of times a block is repeated in Figure 6, we notice that a single repetition already yields the biggest improvement, two repetitions usually achieve the highest accuracy, and looping more times does not lead to additional gains but often results in a decrease.

## 5. Adaptability of Looped and Grown Models

In this section, we study how looped and depth-grown transformers adapt. We first consider simple adaptation settings (§5.1), few-shot in-context learning and supervised fine-tuning on reasoning primitives, where both looped and grown models improve faster than the baseline. We then move to more complex pre-training settings, higher-quality math cooldown mixtures (§5.2), and retrofitted recurrence (§5.3), showing that the grown model (LIDAS) achieves the largest overall reasoning gains and makes the best use of

additional inference-time repetitions.

### 5.1. In-Context Learning & Supervised Fine-Tuning

**In-Context Learning.** To assess in-context learning, we evaluate each reasoning primitive with an increasing number of examples, up to the context-length limit. In Figure 7, looped and depth-grown models benefit more from additional examples than the baseline, which often shows little to no improvement. With enough examples, Loop (4-4×4-4) sometimes even surpasses the grown model LIDAS, consistent with Geiping et al. (2025), who observe that dynamically trained looped models make better use of additional in-context examples at higher recursion. In general, not all reasoning primitives benefit from more examples, see Appendix C.2.

**Supervised Fine-Tuning.** To evaluate supervised fine-tuning, we use the *Variable Assignment Code* task, focusing on the depth-1 ($d = 1$) and depth-2 ($d = 2$) variants with one and two assignment hops, respectively (see Appendix D). This is a challenging setting: without fine-tuning, most models remain near chance. Following Saunshi et al., 2024, we fine-tune on additional examples from the depth-1 and depth-2 variants, using an equal number of samples from each. In Figure 8, with just 64 training examples, LIDAS and Loop (4-4×4-4) already rise above chance, while the baseline remains close to random even with 128 examples. As the dataset grows, all looped and grown models outperform the baseline. Notably, for the depth-2 variant, Loop (4×6) matches LIDAS at larger dataset sizes, while Loop (4-4×4-4) reaches even higher accuracy.

### 5.2. High-Quality Cooldown Mixtures

Moving beyond supervised fine-tuning, we investigate whether the improved adaptability of grown and looped

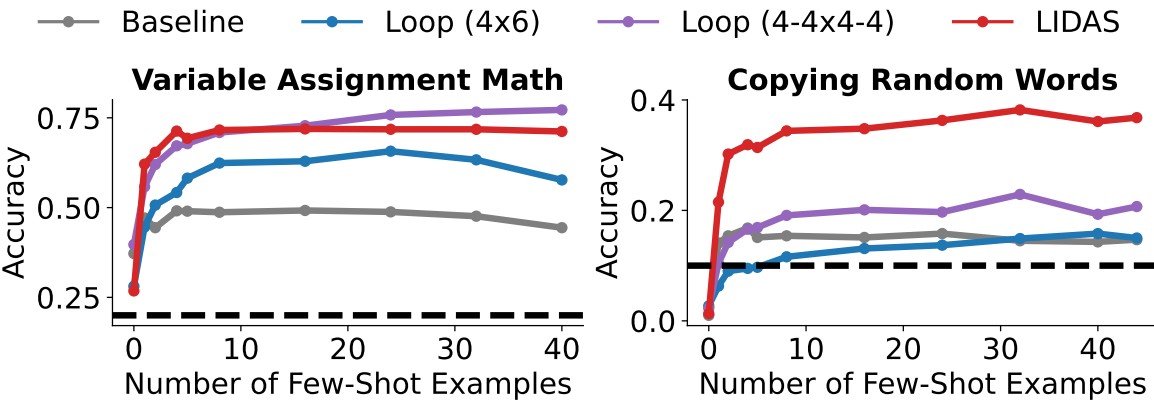

*Figure 7.* **Looped and depth-grown models use in-context examples better.** As we increase the number of in-context examples up to the maximum context length, the grown—and especially the looped—models improve, while the baseline often does not.

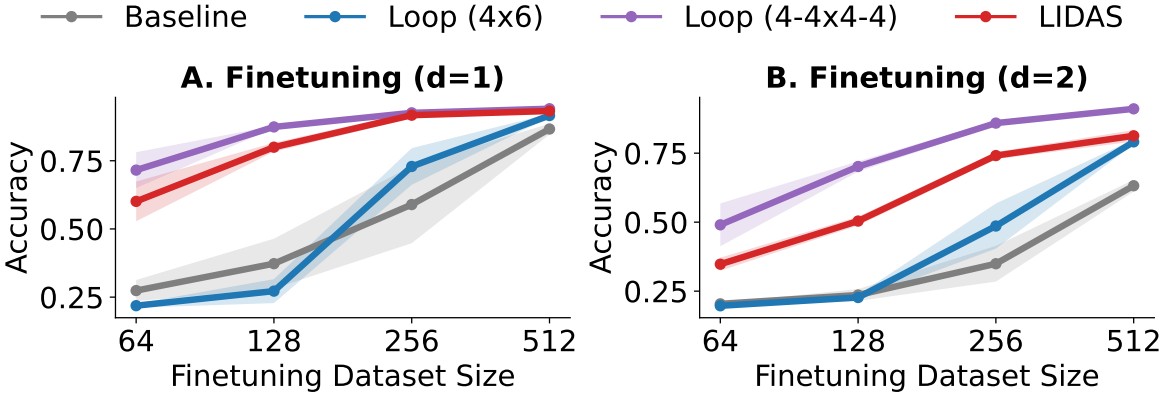

*Figure 8.* **Looped and grown models benefit substantially more from supervised fine-tuning than the baseline.** Shaded regions indicate ± one standard deviation over three random seeds (varying the fine-tuning dataset).

models also holds in a more complex pre-training setting: upsampling high-quality math tokens during the final stage of training (Blakeney et al., 2024). Following commonly used cooldown setups, often referred to as mid-training, and math ratios (Allal et al., 2025; Zhu et al., 2025b; Walsh et al., 2025), we increase the math fraction to 20% during the last 30k steps (15% of pre-training) and ablate the source of math tokens, comparing FineMath-4+ (FMT) (Allal et al., 2025) to Nemotron-CC-Math-4+ (NMT) (Mahabadi et al., 2025). As shown in Table 2, NMT yields larger gains for the 360M models on Math Word Problems, Reasoning Primitives, and GSM8K (Cobbe et al., 2021), and we therefore use it in subsequent experiments. Applying this improved cooldown at 1.7B for the Baseline, LIDAS, and Loop (4-4×4-4), all models improve (Table 3), with LIDAS achieving the strongest gains on the same reasoning benchmarks, and especially GSM8K.

*Table 2.* **Nemotron-CC-Math-4+ leads to highest reasoning gains.** We ablate the effect of increasing the proportion and quality of math tokens (from 6% OpenWebMath in gray) during the cooldown on reasoning performance. Both FineMath-4+ (FMT) and Nemotron-CC-Math-4+ (NMT) increase the performance on Math Word Problems, Reasoning Primitives and GSM8K.

| | | Math Cooldown Dataset | | |
|---|---|---|---|---|
| | | **Math Word Problems** (Acc ↑) | **Reasoning Primitives** (Acc ↑) | **GSM8K** (Acc ↑) |
| 360M | Baseline | 3.69 | 30.04 | 1.06 |
| | + FMT 20% | 13.46 | 30.10 | 2.12 |
| | + NMT 20% | 21.15 | 35.70 | 3.41 |
| | MIDAS | 4.39 | 28.42 | 1.44 |
| | + FMT 20% | 14.68 | 32.84 | 3.49 |
| | + NMT 20% | 23.70 | 38.42 | 4.09 |
| | LIDAS | 4.36 | 31.58 | 1.36 |
| | + FMT 20% | 16.54 | 40.12 | 3.94 |
| | + NMT 20% | 25.64 | 42.20 | 6.07 |

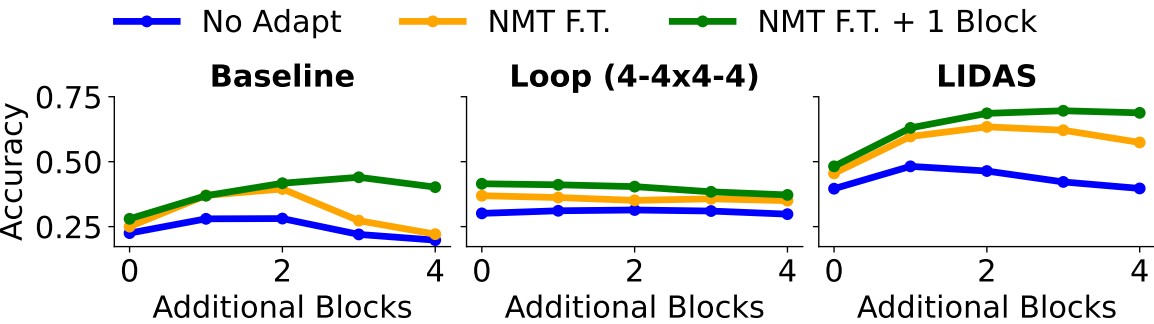

*Figure 9.* **LIDAS benefits the most from additional inference repetitions.** Repeating a fixed 4-layer block at inference increases accuracy on the reasoning primitive *Variable Assignment Basic*. LIDAS and the baseline make the best use of additional blocks, up to 3 repetitions, after adapting the models to loop over that block once. The looped model does not benefit from further repetition of the recurrent block, but it also degrades less than the baseline, especially before the baseline is adapted to looping.

## 5.3. Retrofitted Recurrence

We previously found that depth-grown models can benefit from additional inference compute by looping a middle block. Following Koishekenov et al. (2025), we retrofit this recurrence during cooldown by training on the improved cooldown mixture while looping a selected 4-layer middle block for one additional repetition. Table 3 supports two observations. Leveraging the grown model's natural 4-layer block structure to choose candidate middle blocks, looping the 3rd block (layers 8–11) is the best overall choice for both LIDAS and the baseline, especially on GSM8K, while looping layers 12–15 is slightly weaker overall. In contrast, looping Loop (4-4×4-4)'s recurrent block one additional time yields only modest gains compared to the corresponding improvements for the baseline and LIDAS. Appendix Figure 17 provides the full ablation over the retrofitted block position. Consistent with Figure 6 and prior findings in the literature (Koishekenov et al., 2025), the middle layers appear to play the most substantial role in the latent reasoning process.

Using the third-block-adapted variants, Figure 9 shows that LIDAS benefits the most from repeating the adapted block additional times at inference time. More broadly, this adaptation makes performance changes more stable when running more repetitions than seen during training. Notably, the baseline, which previously showed noticeable degradation with three additional repetitions, can improve slightly after adaptation. LIDAS and the looped model also show less degradation without adaptation. For additional results, see Appendix C.3.

## 6. Conclusion

Depth growing reduces pre-training FLOPs and produces a weight structure closely aligned with looped models, while retaining the flexibility of untied layers. Across architectures, both exhibit convergent depth-wise signatures, greater

*Table 3.* **LIDAS benefits most from math cooldown and retrofitted recurrence.** Baseline, LIDAS, and Loop (4-4×4-4) (1.7B) before (gray) and after math cooldown with 20% Nemotron-CC-Math-4+ (NMT), with and without a single additional loop of a 4-layer block (0-indexed layer range). Best values per model and metric are bolded.

| | | | Math Cooldown ± Loop | | |
| | | Looped Block Index | Math Word Problems (Acc ↑) | Reasoning Primitives (Acc ↑) | GSM8K (Acc ↑) |
|---|---|---|---|---|---|
| | Baseline | - | 13.61 | 34.62 | 2.35 |
| | + NMT 20% | - | 32.99 | 39.20 | 10.61 |
| | + NMT 20% | 8-11 | **34.75** | **40.90** | **12.59** |
| | + NMT 20% | 12-15 | 34.02 | 39.84 | 11.75 |
| **1.7B** | LIDAS | - | 18.18 | 48.02 | 5.61 |
| | + NMT 20% | - | 35.54 | 52.60 | 13.34 |
| | + NMT 20% | 8-11 | **38.84** | 54.84 | **17.36** |
| | + NMT 20% | 12-15 | 38.14 | **56.60** | 15.47 |
| | Loop (4-4×4-4) | - | 6.91 | 39.88 | 2.65 |
| | + NMT 20% | - | 27.00 | 41.94 | 6.52 |
| | + NMT 20% | 4-7 | **27.74** | **43.62** | **7.35** |

reliance on late layers and recurring residual-stream and sublayer-usage patterns around the looped/grown block, that support a shared mechanism for iterative refinement. These parallels suggest that the benefits of growing and looping come from similar repeated computation across depth. This connection makes the techniques adaptable and composable: they adapt better than baselines with more in-context examples or supervised fine-tuning data, and depth-grown models benefit most from higher-quality, math-heavy cooldown mixtures. Depth-grown models can be retrofitted to loop a middle block during cooldown, improving reasoning capabilities. In our setting, this "grow first, loop later" strategy yields the strongest reasoning performance among the standard, looped, and depth-grown models under matched data and inference FLOPs, suggesting a simple reasoning recipe. We hope these findings guide more efficient scaling strategies for future reasoning-capable transformer models accross diverse domains.

## Acknowledgments and Disclosure of Funding

The authors would like to thank Nino Scherrer and Tobias Höppe for insightful discussions and support throughout this work.

This work was partially supported by the Helmholtz Foundation Model Initiative and the Helmholtz Association. The authors gratefully acknowledge the Gauss Centre for Supercomputing e.V. (www.gauss-centre.eu) for funding this project by providing computing time through the John von Neumann Institute for Computing (NIC) on the GCS Supercomputer JUPITER — JUWELS (Jülich Supercomputing Centre, 2021) at Jülich Supercomputing Centre (JSC). Furthermore, the authors appreciate the computational resources provided by the National High Performance Computing Centre (www.nhr.kit.edu). The research presented is supported by the TUM Georg Nemetschek Institute Artificial Intelligence for the Built World and the German Federal Ministry of Education and Research (Grant:01IS24082).

## Impact Statement

This paper presents work whose goal is to advance the field of Machine Learning. There are many potential societal consequences of our work, none which we feel must be specifically highlighted here.

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

# A. Details of Growing

We briefly summarize the training procedure used to obtain the depth-grown models MIDAS and LIDAS in this work. The implementation follows the growing methodology introduced by Saunshi et al. (2024) and analyzed in detail by Kapl et al. (2025); we refer the reader to those works for a complete description.

We use a fixed block size $B = 4$ and perform gradual depth expansion by inserting a new *middle* block after each stage. Depending on the variant, this corresponds to MIDAS (duplicating the middle block of the current stage) or LIDAS (duplicating the layer-wise middle). Model width and the number of attention heads remain unchanged throughout growth.

At each growth step, layer parameters and their optimizer state are deep-copied so that duplicated layers start from identical weights and AdamW moments, and then diverge through continued training. Token embeddings and the final output head are copied without modification.

Let $L_{\text{final}}$ denote the final depth and $k = L_{\text{final}}/B$ the number of growth stages. If $T$ is the total number of training steps, the budget for stage $i$ is allocated using the PROP-$\alpha$ schedule of Saunshi et al. (2024):

$$T_i = \frac{i^\alpha}{\sum_{j=1}^{k} j^\alpha} T \qquad i = 1, \ldots, k$$

and we use PROP-1 ($\alpha = 1$) in all experiments. In practice, $T_i$ is rounded to integers while preserving $\sum_i T_i = T$, and a single continuous learning-rate schedule is maintained across stages (no learning-rate reset). We set $T = 170,000$ so that all models reach their final depth before entering the cooldown phase.

To complement the main trade-off summary in Figure 1, which focuses on *Reasoning Primitives*, Figure 10 breaks down the same parameter/inference/training trade-offs by benchmark category from Table 1.

# B. Additional Mechanistic Analysis

This section provides additional mechanistic analyses that complement the results presented in Section 4.1. The experiments here further probe how architectural repetition patterns manifest in intervention robustness, block reuse, and sublayer-level behavior across models.

## B.1. Swapping Interventions

We present an additional swap experiment in Figure 11, complementing the block-size-1 results shown in Figure 5. Consistent with those findings, looped models again exhibit lower robustness to structural interventions.

When swapping a consecutive block of two layers, the fully looped model Loop (4×6) shows a pronounced degradation in performance on both Lambada and Variable Assignment (Math) compared to the baseline, LIDAS, and the partially looped variant Loop (4-4×4-4).

## B.2. Repeat Block experiments

This section complements Section 4.2 and Figure 6 by providing additional plots for the *repeat-block without adaptation* setting across the rest reasoning primitives (Figure 12).

Consistent with the main findings, repeating a contiguous block of four layers located in the middle of the network (starting around layer 10), without any further training, leads to measurable performance gains on the reasoning primitives tasks, although for some tasks is less pronounced.

## B.3. Sublayer Usage Experiments

In this section, we provide additional plots analyzing sublayer usage, complementing Section 4.1.

First, Figure 13, complementing Figure 4, shows that both looped models exhibit a periodic pattern in which specific layers are highly sensitive to changes in all preceding layers. This behavior closely resembles that of LIDAS and contrasts with the Baseline, where no clear structure emerges.

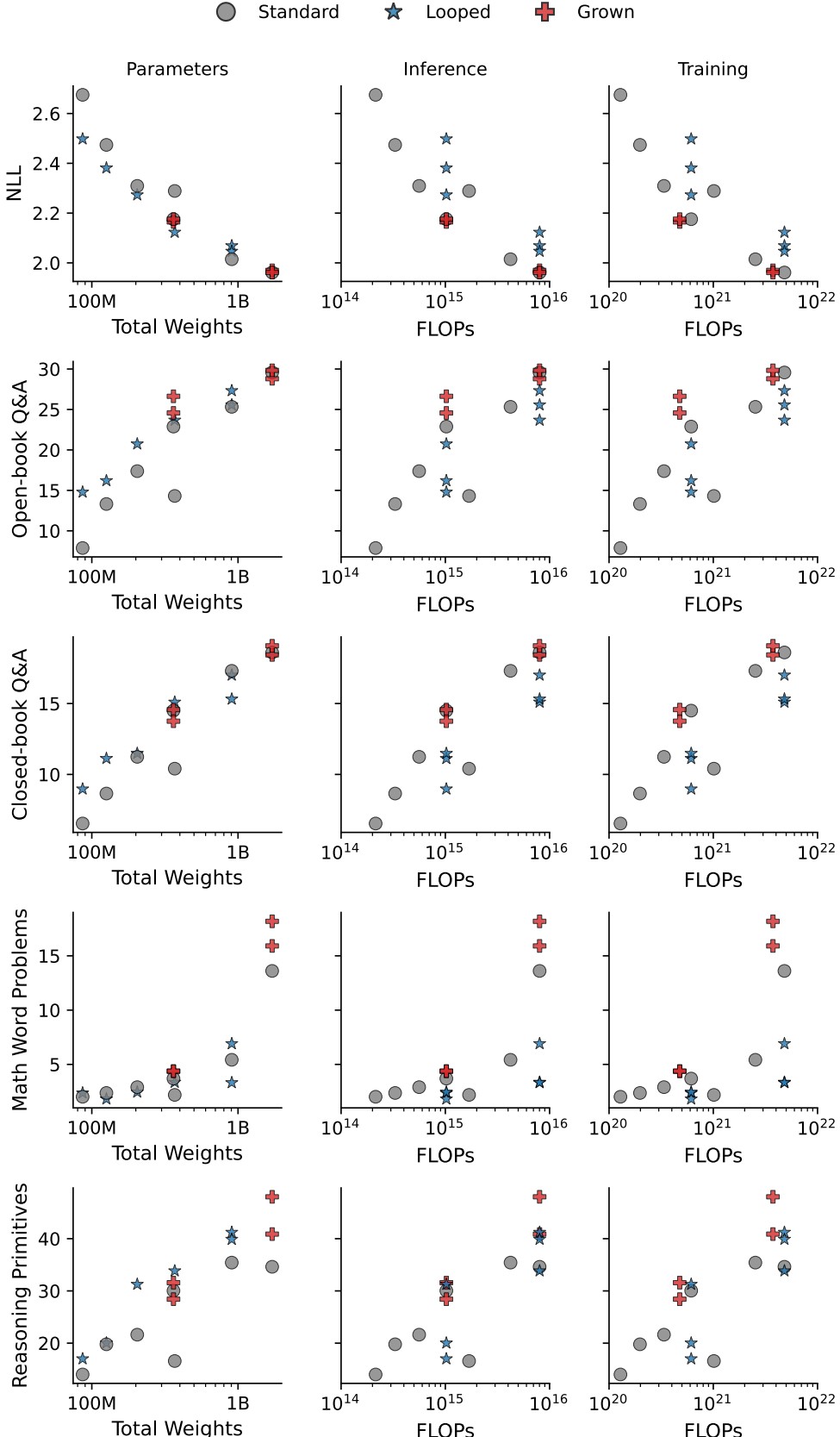

*Figure 10.* **Category-wise trade-offs for looped and depth-grown models.** Each point corresponds to a model in Table 1 (up to 1.7B parameters). For each benchmark category, we plot the average metric versus unique parameters (left), inference FLOPs (middle), and training FLOPs (right), complementing Figure 1.

Next, Figure 14, complementing Figure 3, reports the relative contribution of each layer (Attention + MLP) with respect to its input. Around the middle of the network, periodic local maxima remain visible, primarily for LIDAS and Loop (4×6), recurring every four layers.

## C. Additional Adaptability and Inference Scaling Results

This section provides additional results that complement the adaptability and inference-scaling analyses presented in the main text. We further examine how looped and depth-grown models respond to supervised fine-tuning and to inference-time repetition across tasks, highlighting how their architectural inductive biases continue to influence performance beyond the original training setup.

### C.1. Supervised Fine Tuning

In Figure 15, we present an extended version of the results shown in Figure 8. Beyond the fine-tuning behavior on the Depth 0 variant of the Variable Assignment task, we additionally include two hybrid models: variants of Loop (4-4×4-4) and Loop (4×6) in which weight tying is removed during the fine-tuning phase.

Despite having strictly more degrees of freedom during adaptation, these hybrid variants consistently underperform their original tied counterparts across all depths. This suggests that the inductive bias introduced by weight tying continues to play a beneficial role even during supervised fine-tuning.

### C.2. In-Context Learning

In Figure 16, we show the In Context Learning Behaviour for all reasoning primitives tasks, complementing Section 5.1.

### C.3. Inference Scaling

In Figures 18 and 19, we present additional results on *retrofitted recurrence* across all reasoning primitives and GSM8K, complementing Figure 9 in the main text. These plots confirm the same trend observed there: introducing inference-time repetition in the middle blocks consistently improves performance for the depth-grown models.

Additionally, we ablate which 4-layer block is retrofitted to loop during cooldown. Figure 17 shows that the strongest candidates are typically in the middle of the network: among the six 4-layer blocks, the third (layers 8–11) and fourth (layers 12–15) blocks are consistently competitive, with the third often strongest on reasoning-heavy tasks. This motivates our default choice of adapting the third block in the main experiments.

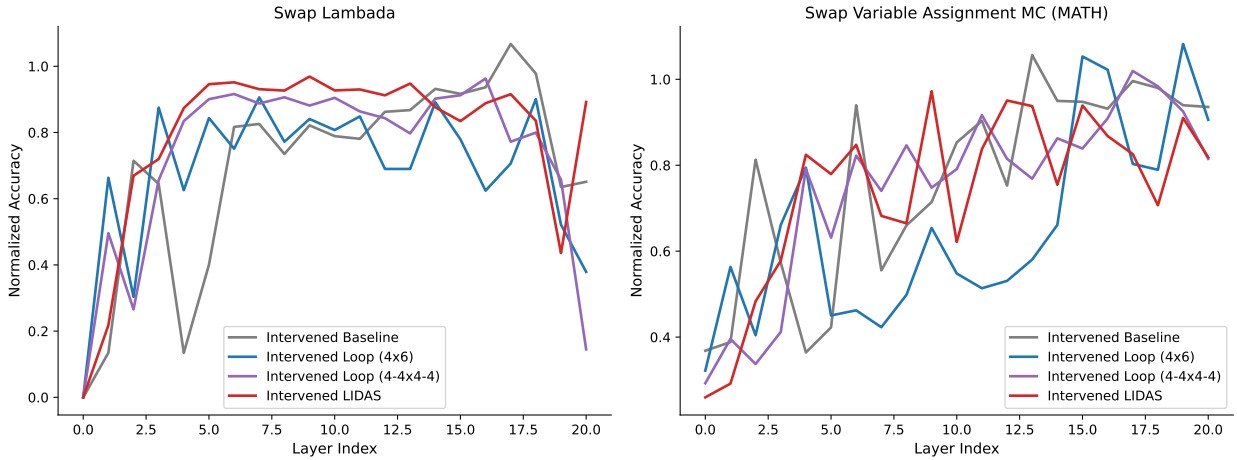

*Figure 11.* Swapping consecutive 2-layer subblocks

# D. Tasks and Benchmarks Overview

## D.1. Reasoning Primitives

We implement the **Reasoning Primitives** tasks according to the setup introduced by Saunshi et al. (2024).

The copying task is constructed by first sampling a sequence of random three-letter tokens (e.g., a sequence of length 10). A contiguous segment of this sequence (e.g., length 5) is then repeated at the end, and the model is asked to predict the next token in the original order. This formulation isolates the ability to track sequence structure and positional dependencies without relying on semantic cues.

An illustrative example of the *Copying random words* task is:

*Prompt*:

```
Fill in the blank:
jic dqy sof uzg ewr oxw osp tkj rvw mnu jic dqy sof uzg ewr ___. ->
```

*Answer*:

```
oxw
```

The variable assignment task is constructed by sampling a collection of variable–value bindings and asking for the value of a queried variable after the assignments have been processed. We use the same prompt templates for the *basic*, *math*, and *code* variants of this task.

An important notion in this task family is the **depth**, which controls how many intermediate substitutions are required before the queried variable can be resolved. At depth $d = 0$, the queried variable appears directly among the assignments. At higher depths, the queried variable must be resolved through a chain of dependencies, requiring the model to iteratively propagate values across multiple steps.

An example of the *Variable assignment* task at **depth 0** (Basic) is:

*Prompt*:

```
Fill in blank:

o=23
k=3
t=13
a=1
e=9
o=___. ->
```

*Answer*:

```
23
```

An example at **depth 1** is:

*Prompt*:

```
Fill in blank:

o=2
k=23
t=13
a=1
e=9
v=k
```

```
c=e
s=o
y=t
r=a
y=___.  ->
```

*Answer*:

```
13
```

Here, the model must first resolve the value of `t` before determining the value of `y`.

An example at **depth 2** is:

*Prompt*:

```
Fill in blank:

o=2
k=23
t=13
a=1
e=9
v=k
c=e
s=o
y=t
r=a
b=r
h=c
f=y
x=s
g=v
h=___.  ->
```

*Answer*:

```
9
```

In this case, correctly answering requires following a two-step chain of substitutions, illustrating how increasing depth demands progressively longer reasoning chains.

All evaluations are conducted in a multiple-choice setting with five in-context examples. Under this protocol, random guessing corresponds to a baseline accuracy of 10 for the copying task and 20 for the variable assignment task.

## E. Experimental Protocol for Depth and Mechanistic Analysis

**Codebase and methodology.** Our analysis follows the intervention framework introduced by Csordás et al. (2025), adapted to the setting of our models. We perform single-layer interventions to quantify how information introduced at one layer affects representations in later layers. Tuned Lens experiments follow the procedure of Belrose et al. (2023).

**Models and evaluation data.** All analyses are conducted on SmolLM backbones at 360M and 1.7B parameters (Ben Allal et al., 2024; Kapl et al., 2025). For consistency across experiments, we use the same fixed set of GSM8K prompts and evaluation settings throughout all depth and early-exit analyses.

**Future local effects.** To measure how information propagates forward through the network, we use the *future local effects* protocol. For a given source layer $s$ and a later layer $\ell > s$, we remove the contribution of layer $s$ from the residual stream

that is fed into layer $\ell$, while keeping the rest of the forward pass unchanged. We then measure the relative change induced at layer $\ell$ compared to the original, unmodified forward pass.

This procedure isolates the direct influence of layer $s$ on layer $\ell$ without allowing the modification to propagate further through the network. Repeating this for all pairs $(s, \ell)$ yields a matrix of relative changes that forms the basis of the heatmaps shown in the appendix.

The relative change is computed as the norm of the difference in the residual update at layer $\ell$ divided by the norm of the original residual update. For visualization, we aggregate these values by taking the maximum across batch examples and sequence positions, resulting in a single source–target matrix per model.

**Tuned Lens early-exit evaluation.**   For early-exit analyses, we train a small affine adapter for each layer that maps its residual output to the representation space immediately before the final normalization and unembedding layer, following Belrose et al. (2023). Logits are then obtained by applying the model's final normalization and unembedding.

Adapters are trained on a held-out split of FineWeb-Edu. We evaluate early-exit quality by measuring (i) the KL divergence between the early-exit and final output distributions, and (ii) the overlap of the top-5 predicted tokens with those of the final layer.

**Depth score.**   To summarize how strongly different layers influence the model's output, we compute a depth score based on the change in the output distribution when intervening at each layer. For each layer, we measure the maximum L2 change in the softmaxed logits caused by the intervention, average this quantity across examples, normalize the resulting vector across layers to form a distribution, and report its expected layer index as the depth score, following Csordás et al. (2025).

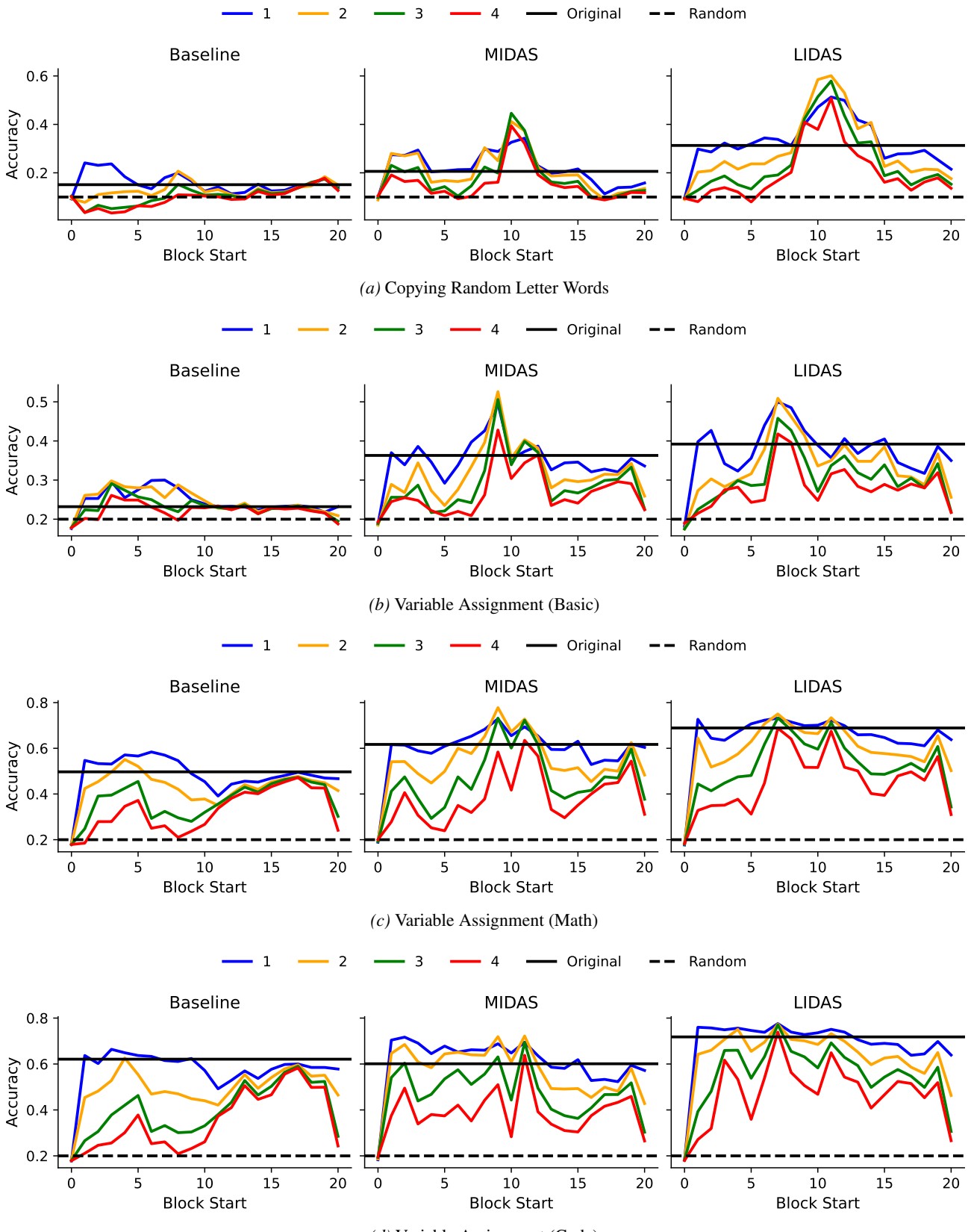

*Figure 12.* Repeat-block ablations across reasoning primitive tasks without further training.

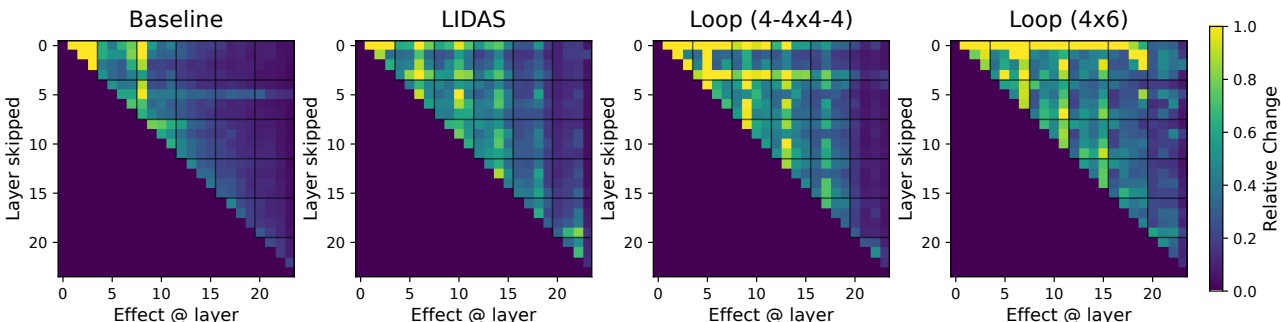

*Figure 13.* Future Local Effects

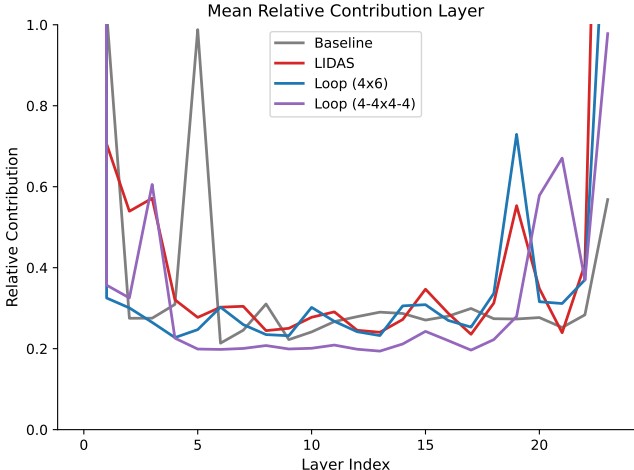

*Figure 14.* Mean Relative Layer Contribution

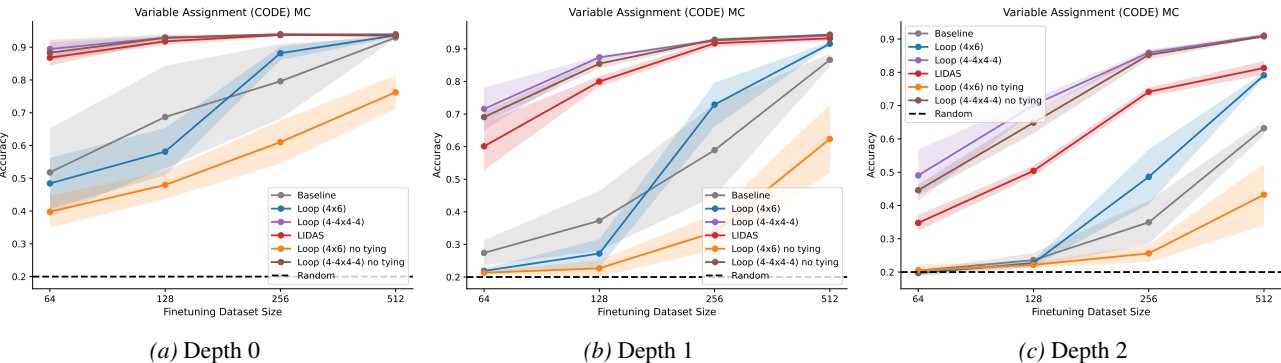

*(a)* Depth 0        *(b)* Depth 1        *(c)* Depth 2

*Figure 15.* Supervised Fine Tuning Experiments

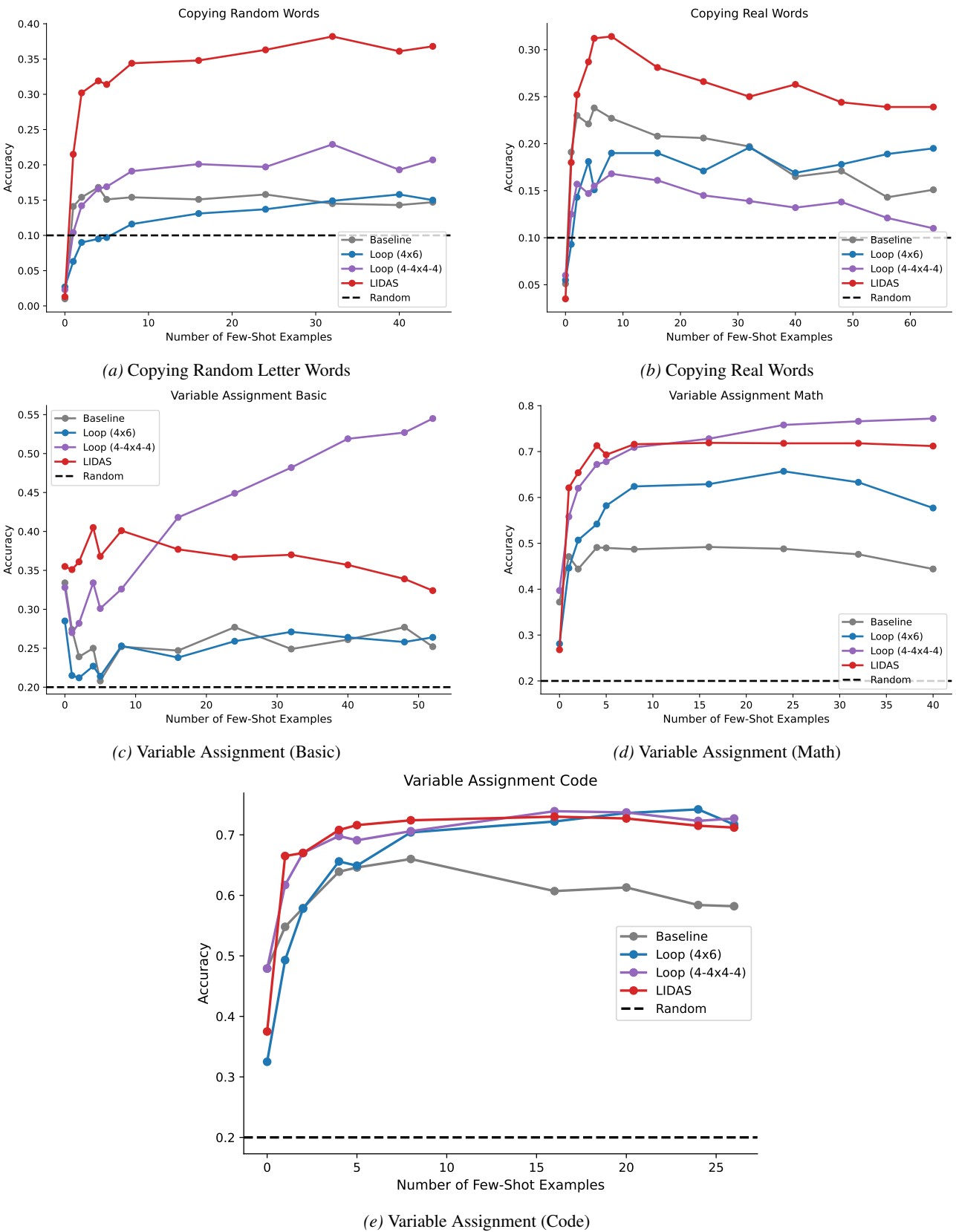

*Figure 16.* In-context learning behavior across all reasoning primitive tasks.

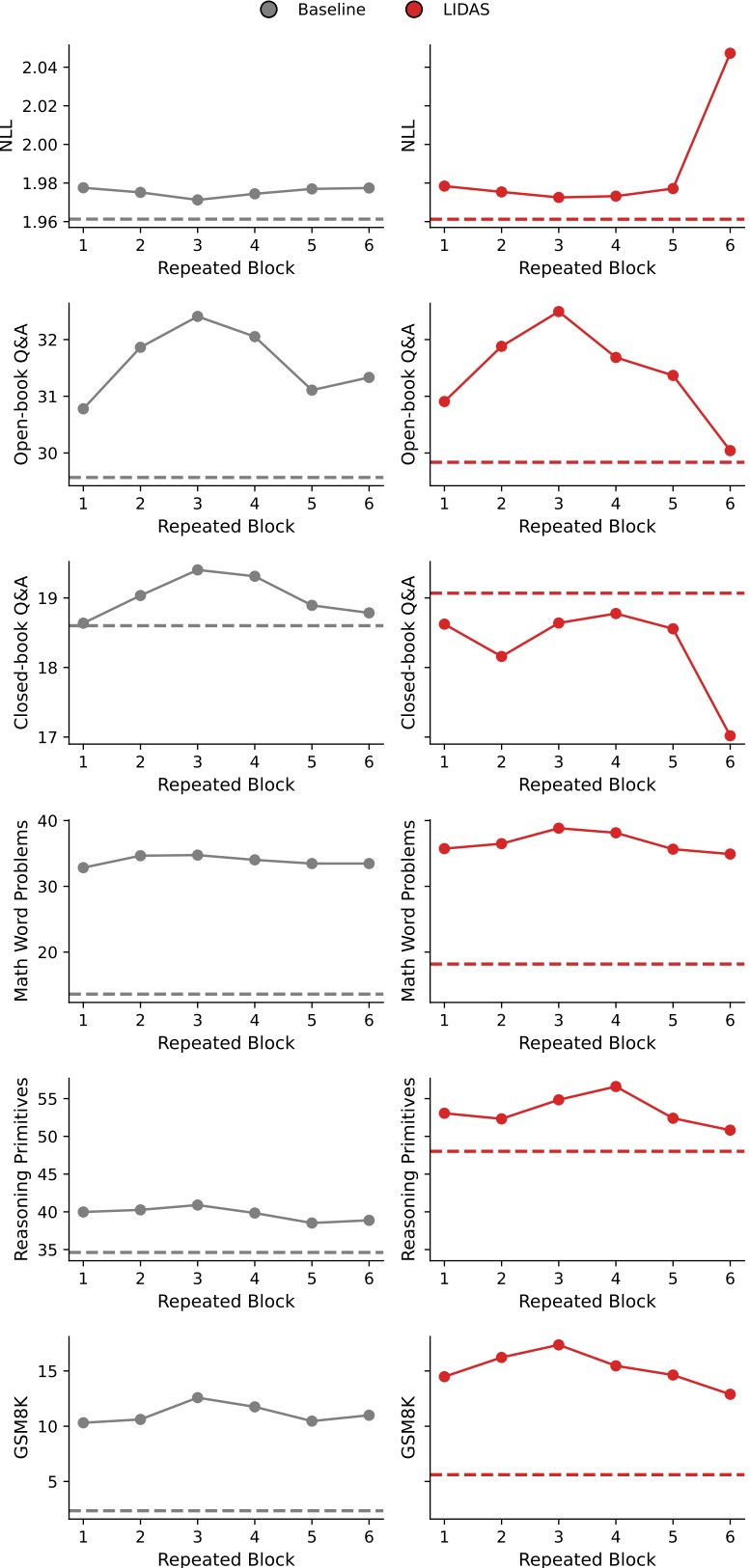

*Figure 17.* **Retrofitted-recurrence block position ablation.** We sweep which 4-layer block is looped once during cooldown adaptation for the baseline and LIDAS. Dashed curves show performance before adaptation, and solid curves show performance after adaptation of the respective block. Middle blocks, in particular the third (layers 8–11), tend to be the strongest candidates, especially for reasoning.

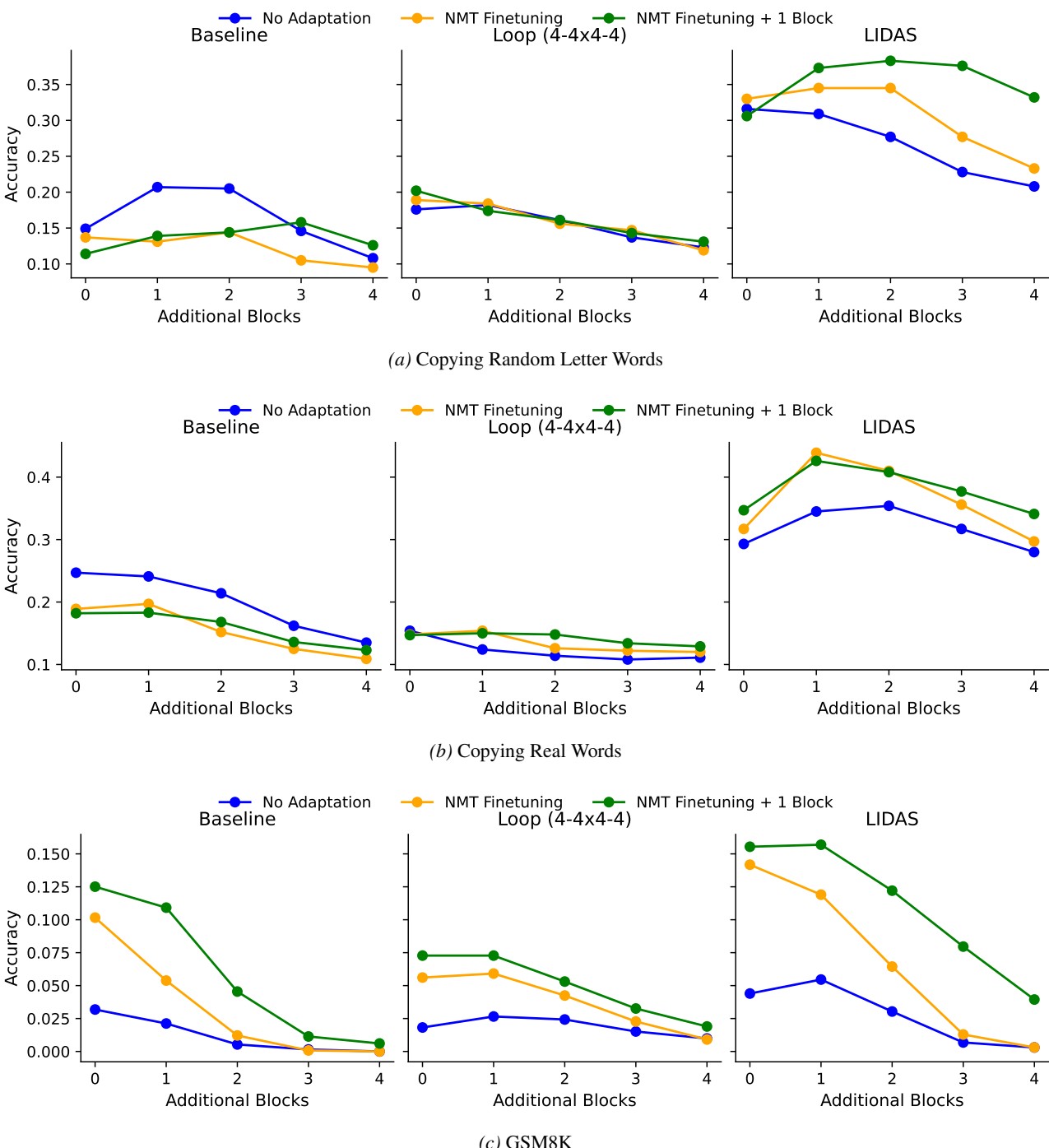

*(a)* Copying Random Letter Words

*(b)* Copying Real Words

*(c)* GSM8K

*Figure 18.* Repeating blocks without further training (NMT finetuning setting) on copying tasks and GSM8K.

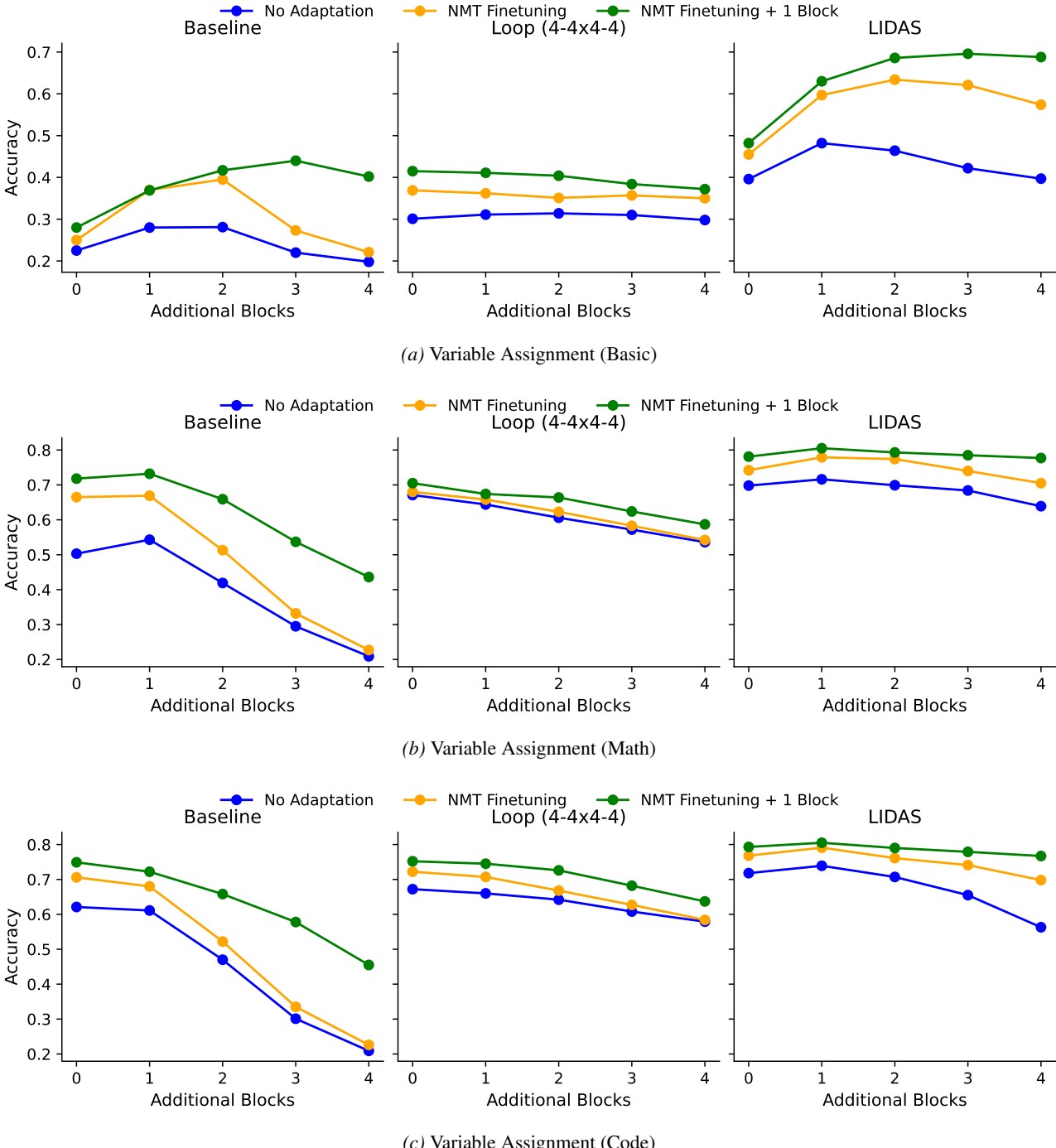

*(a)* Variable Assignment (Basic)

*(b)* Variable Assignment (Math)

*(c)* Variable Assignment (Code)

*Figure 19.* Repeating blocks without further training (NMT finetuning setting) on Variable Assignment tasks.

