# OpenReview forum: "From Growing to Looping: A Unified View of Iterative Computation in LLMs"
_ICML.cc/2026/Conference — ICML 2026 regular_

### Official Review · Reviewer_Crin · 2026-03-11

**Soundness:** 3
**Presentation:** 4
**Significance:** 4
**Originality:** 3
**Overall Recommendation:** 6
**Confidence:** 4

**Summary:**

This paper presents a systematic comparative study of two Transformer variants that both rely on iterative computation and share partially similar structural intuitions: Looped Transformers and Grown Trained Transformers. The paper compares them along three main axes: (1) performance relative to standard models under matched compute and matched parameter budgets, (2) mechanistic similarities and differences, including effective depth, robustness, and periodicity in internal activations, and (3) the possibility of combining the two paradigms, both without additional training and through mid-training interventions. Overall, the paper offers a careful comparative perspective on the commonalities and distinctions between these two model families, provides fairly deep mechanistic analysis, and points to a potentially promising direction: further looped extension on top of grown models.

**Compliance With Llm Reviewing Policy:**

Affirmed.

**Final Justification:**

I have scored it as 6 in the initial review. As I have expected, many of my questions cannot be answered in rebuttal. I believe it is a good paper.

**Key Questions For Authors:**

The following is some questions related to the paper but not the flaw of it.

1. Figure 2 seems to suggest that, in terms of the signature-related behavior, the looped model lies *between* the grown model and the baseline. This is somewhat surprising, since intuitively I would have expected the looped model to be the more extreme case relative to the grown model. From my intuition, the grown models seems a looped initialization and then finetuned to be more "normal". Do the authors have a more detailed explanation for this? One possible interpretation is that the additional training of the grown model further increases usage of later layers beyond what is induced by initialization alone, but this would benefit from a more explicit discussion.
2. Is depth curse actually harmful? From the paper’s results, it is not obvious that this phenomenon is harmful from a performance perspective. Looped Transformers appear to improve effective depth, yet under matched inference budget they can underperform the baseline; similarly, Grown models only show relatively modest gains. This raises the possibility that the depth curse may not simply be a pathology, but in some settings may even be tied to strong performance. Alternatively, current methods for increasing effective depth may partly work by weakening earlier layers and forcing the model to use later layers, rather than by creating uniformly better computation. This is a large question, but I think the discussion section would benefit from engaging with it more directly.
3. The paper states that *"the middle of the network as a relaxed version of a fully looped or tied model."* I would appreciate a more precise explanation of this claim. How exactly is this conclusion derived from the experiments? What structural or functional evidence supports the “relaxed” interpretation? More importantly, what does this suggest for improving reasoning performance in practice? This statement feels potentially important, but its implications are not yet fully unpacked.
4. One of the most interesting findings is that a grown model can be additionally looped at inference time even if it was not explicitly trained for that setup, whereas the baseline model cannot. This seems like a genuinely exciting result that may open up a new direction for inference-time scaling. At the same time, the effect appears less clear on *mathematical* tasks. So why? I would strongly encourage the authors, either in this paper or in future work, to further explore this direction. Even carefully documented negative results would be valuable here, given how novel and potentially important this avenue seems.

**Limitations:**

I encourage the authors to provide a more clearn limitation or future work section. Some possible limitation has been suggested above.

**Strengths And Weaknesses:**

**Strengths**:
1. **Comparative perspective with mechanistic depth**: The major strength is the comprehensive experiments, with well organization, clear explanation and convicing conclusion.
The analyses of effective depth, robustness, and periodicity of internal activations are interesting
2. **Promising hybrid direction**: The paper usefully demonstrates the possibility of combining growing and looping, including both training-aware and training-free settings. In particular, the observation that a grown model can sometimes be further looped at inference time, even without dedicated training for that regime, is inspired and suggests a potentially new direction for inference-time scaling.
3. **Clear framing and presentation**: The paper is very clearly written. The motivation is easy to follow, the related work is well organized, and the notation is introduced in a clean and readable way. This significantly improves accessibilit.

**Weaknesses**:
1. In the main Table 1, the gap between LIDAS and the baseline is often within 1%. This makes some of the empirical conclusions fell elss decisive than the paper's framing might suggest. Though it is expensive, if the authors can train the models with several times and report the variance, the results will be more convincing. (Note that the variance should be the re-trained variance, instead of the test-time variance.)
2. **Overly dense scope**: While it is not a flaw, the paper contains a large amount of information. The current presentation feels somewhat compressed by the page limit. Some sections would likely benefit from being expanded into a separate paper with more thorough experiments and fuller discussion. As a result, the current manuscript can at times feel crowded relative to the breadth of its ambitions.
3. **Minor presentation issues**: There are also a couple of small presentation problems. First, I would expect Table 1 to appear on page 4, where the corresponding analysis is discussed, rather than earlier. Second, Table 1 appears to contain a typo: the last standard model is labeled as 12/24, which seems incorrect, since 12/24 does not appear to be a standard setting in this context.

---

> ### Author Rebuttal · Authors · 2026-03-31
>
> We sincerely thank Reviewer Crin for their thorough and thoughtful evaluation of our work. We are very grateful for the recognition of its merits, like the comparative perspective with mechanistic depth, the promising hybrid direction combining growing and looping, and the clarity of our framing and presentation. We are also deeply appreciative of the insightful questions raised and glad for the opportunity to discuss them in more detail.
>
> ## Small performance gaps / lack of variance reporting
>
> We thank the reviewer for this suggestion. We have now run 3-seed retraining experiments (varying data and initialization) for the 360M models and report mean ± std across runs. The results confirm that the main conclusions are robust: LIDAS remains the strongest overall and maintains clear gains over the baseline across several benchmarks, including Reasoning Primitives (35.48 ± 3.39 vs. 30.59 ± 1.06).
>
> | Model | NLL | Open-book Q&A | Closed-book Q&A | Lambada | HellaSwag | Math Word Problems | Reasoning Primitives |
> | :--- | :---: | :---: | :---: | :---: | :---: | :---: | :---: |
> | Baseline | 2.175 ± 0.001 | 23.33 ± 0.54 | 14.95 ± 1.04 | 43.17 ± 0.18 | 39.95 ± 0.22 | 3.20 ± 0.45 | 30.59 ± 1.06 |
> | MIDAS | 2.176 ± 0.002 | 25.29 ± 1.52 | 13.97 ± 0.32 | 43.22 ± 0.63 | 40.37 ± 0.26 | 3.78 ± 0.53 | 30.28 ± 3.29 |
> | LIDAS | 2.164 ± 0.001 | 27.38 ± 2.82 | 14.40 ± 0.28 | 44.16 ± 0.12 | 40.76 ± 0.16 | 3.41 ± 0.89 | 35.48 ± 3.39 |
> | Standard 16 | 2.312 ± 0.002 | 18.09 ± 0.92 | 12.03 ± 0.82 | 37.62 ± 0.39 | 35.85 ± 0.12 | 2.61 ± 0.26 | 23.84 ± 2.43 |
> | Loop (16x2) | 2.272 ± 0.001 | 20.16 ± 0.51 | 11.70 ± 0.28 | 38.49 ± 0.49 | 37.45 ± 0.19 | 2.44 ± 0.03 | 29.01 ± 3.89 |
>
> ## Overly dense scope
> We thank the reviewer for this feedback and will take it into account.
>
>
>
> ## Minor presentation issues
> We thank the reviewer for pointing this out. We will move Table 1 to the appropriate section (page 4) and correct the typo in the final version.
>
> ## Q1: Figure 2 - Why does the looped model not appear as the more extreme case?
>
> We thank the reviewer for this interesting observation. We would like to clarify that while the depth score (Fig. 2a) does indeed place the grown model between the two extremes of the looped model and the fully untied baseline, this pattern does not hold for the Early Exit metrics assessed on the Variable Assignment task (Fig. 2b,c). We are currently conducting a more fine-grained analysis to identify which attention heads are responsible for moving the necessary information to the final prompt position, where the answer token is decoded. Note that the following is preliminary work. For the baseline models, consistent with prior literature, the most sensitive heads are located slightly past the middle of the network (see the jump at layer 13 in Fig. 2c). For the LIDAS model, the picture is more nuanced: at an earlier growth stage (e.g., stage 3 out of 6), these heads are similarly located just past the middle (e.g., at layer 7 out of 12). However, as the network is expanded, the very same heads, still responsible for this information transfer, are pushed toward the later layers (layer 7 out of 12 becomes layer 19 out of 24; see the corresponding jump for LIDAS at layer 19 in Fig. 2c). For the looped model, by contrast, the sensitive heads are distributed across various layers, which may explain the less concentrated signature observed in the figure.
>
> ## Q2: Is the depth curse actually harmful?
> We thank the reviewer for raising this important question regarding the relationship between depth usage and performance. For the comparison between baseline and looped models under an iso-FLOP budget, we hypothesize that the baseline's superior performance is primarily attributable to its greater parameter capacity, rather than to any benefit conferred by lower effective depth. For LIDAS, our current hypothesis is that the increased depth usage is largely a result of the growing process itself (see our response to Q1): as the network is expanded, the attention heads responsible for information transfer are pushed toward later layers, which mechanically increases the depth usage metric. As for the very general question of the curse of depth's harmfulness, we believe it is an open problem and will require further work to clarify.
>
> ## Q3: Clarification on the "relaxed version of a looped model" interpretation
> We refer the reviewer to our first answer to Reviewer rAxJ, where we provide a detailed explanation of this claim and supporting empirical evidence.
>
> ## Q4: Inference-time looping of grown models and mathematical tasks
> We thank the reviewer for this insightful suggestion and share the enthusiasm for this direction. We plan to further investigate the inference-time looping of grown models, including a more thorough analysis of mathematical tasks, in future work.

---

> > ### Author Rebuttal · Reviewer_Crin · 2026-04-01
> >
> > I have scored it as 6 in the initial review. As I have expected, many of my questions cannot be answered in rebuttal. I believe it is a good paper.

---

### Official Review · Reviewer_wjZC · 2026-03-13

**Soundness:** 2
**Presentation:** 1
**Significance:** 2
**Originality:** 3
**Overall Recommendation:** 2
**Confidence:** 4

**Summary:**

The paper draws a connection between looped LMs and grown models and analyzes their similarities through several experiments. The authors also identify that model growing is beneficial for looped LM inference.

**Compliance With Llm Reviewing Policy:**

Affirmed.

**Final Justification:**

After reviewing the work carefully, I believe that it is premature for publication.

Pros: The authors find that model growing introduces an inductive bias toward looping at inference time.

Cons: As other reviewers stated as well, the current form of the paper is overwhelming and requires a significant edit to reduce the number of experiments and provide a clear, concise manuscript. In my opinion, many experiments do not strengthen the narrative and should be removed (see weaknesses 2 and 5).

In addition, the selection of the architecture is non-standard, and it is not clear whether some of the noisier results are a consequence of that choice.

In their rebuttal, the authors did not address my concerns - for example, they did not suggest any edits. Instead, they used more text to justify why all of the experiments should remain in the manuscript.

Since the paper remains unclear and hard to read, I recommend rejecting it in the current round.

**Key Questions For Authors:**

1. In section 4 you show that depth-grown models benefit from inference-time looping. However, this is shown only for one model family. Do the results of this experiment reproduce when tested on a different model family?

2. The paper lacks coherence and many experiments do not seem justified. Can you explain the main goal of the paper, the motivation for each experiment and how it relates to the main goal?

**Limitations:**

yes

**Strengths And Weaknesses:**

**Strengths:**

1. The authors identify that depth-grown models non-trivially benefit from inference time looping.
2. The authors draw similarities between two seemingly different architectures.



**Weaknesses:**
1. The biggest weakness of the paper is that it lacks coherence, making it hard to evaluate. \
Sections 3-5 each introduce new experimental setups (mechanistic diagnostics, intervention robustness, in-context learning, cooldown mixtures, retrofitted recurrence) but the connection between them is unclear. \
E.g., Section 4.1's residual stream analysis does not inform the design choices in Section 5, and the cooldown mixture ablation (Section 5.2) seems disconnected from the mechanistic analysis. The paper would benefit from a focused set of experiments that build on each other toward a clear conclusion, rather than a broad survey of loosely related findings.


2. Contribution 1, "Empirical trade-offs" (L86-94), is far below the threshold required to be considered a valid contribution. \
Contribution 1 is discussed in Section 3. The section does not contain a direct comparison between looped and grown models - the two families are analyzed independently, and the conclusions for each mirror those of prior work [1,2,3], at equal or comparable model scales:
- looping is better when comparing iso-param models [1]
- looping is worse on language modeling and knowledge-intensive tasks w.r.t iso-flop models [1]
- depth-grown models improve reasoning while preserving broad capabilities [2]

However, it should be noted that the previous works were cited properly.

3. In a similar manner, figure 1 repackages the information in table 1 and again reaches conclusions mentioned in previous works [1,2] (growing improves reasoning and saves training FLOPs [2], etc.).

4. The evaluation results seem noisy, and some conclusions are based on unclear trends. \
E.g. in Figure 1 the iso-inference plot (middle) sometimes shows that looped lms are worse and sometimes better than the standard llm, and that grown may perform the same or may perform much better. A possible explanation is the authors' unconventional architecture selection (the 1.7B model has only 24 layers while the 360M model has 32 - which diverges much from the widely used GPT3 scaling spec [4]). \
Another example is in L214-218: "when retaining enough unique parameters, roughly 50% of the baseline, e.g. Loop (16×2) or Loop (12×2) for the 360M and 1.7B model, respectively, looped models exceed baseline accuracy on reasoning primitives." This result contradicts previous findings in [1], which show that even 1/6 of the parameters are sufficient for looped models to surpass the baseline on reasoning primitives.

5. Section 4.1 - Figure 2 (middle, right): The Tuned Lens diagnostics do not clearly support the claim that looped and grown models share similar depth-wise signatures. In the Top-5 Overlap plot (middle), the differences between models are subtle and all curves follow a similar trend. In the Early Exit plot (right), the looped models track closer to the baseline than to LIDAS. This undermines the "unified signatures" narrative - the strongest signal comes from LIDAS alone, not from a shared looped+grown pattern.


**Minor Weakness:**
1. Table 1 - please add a description that explains the Params/FLOPs column

. \
**References:** \
[1] Reasoning with Latent Thoughts: On the Power of Looped Transformers; Sanushi et. al, ICLR 2025 \
[2] On the Inductive Bias of Stacking Towards Improving Reasoning; Sanushi et. al, NeurIPS 2024 \
[3] Do Depth-Grown Models Overcome the Curse of Depth? An In-Depth Analysis; Kapl et. al, Arxiv 2025 \
[4] Language Models are Few-Shot Learners; Brown et. al, NeurIPS 2020

---

> ### Author Rebuttal · Authors · 2026-03-31
>
> We thank the reviewer for the detailed reading. We address each concern below.
>
> ### W1 / Q2: Paper lacks coherence
> We appreciate this feedback and agree that making the connecting thread more explicit would help. The overall goal of the paper is to **unify, understand, and extend** previous observations about looped and depth-grown models. Each section is motivated by a concrete question:
>
> 1. **Section 3 (Empirical trade-offs):** *Do looped and grown models share an inductive bias toward reasoning?* The key finding is that growing shifts the Pareto frontier of reasoning toward lower training FLOPs, while looping provides a complementary trade-off at fixed inference budgets. This motivates the next question: do these gains arise from a shared mechanism?
> 2. **Section 4 (Mechanistic analysis + inference-time looping):** *What computational signatures do they share?* We use depth-utilization diagnostics, residual-stream analysis, and layer interventions (Section 4.1) to reveal shared signatures. Section 4.2 directly tests an implication of these signatures: if the middle of a grown model functions like a looped block, it should be amenable to inference-time looping without any training change.
> 3. **Section 5 (Adaptability + composability):** *Can we make practical use of this connection?* Prior work has shown initial evidence that looped models benefit from more ICL examples (Geiping et al. 2025) and that depth-grown models adapt better via SFT and mid-training [2, 3], but always in isolation. Here, we compare them directly on ICL, SFT, and high-quality cooldown training. Finally, retrofitted recurrence shows how to make the composition explicit during training to further boost reasoning.
>
> ### W2 / W3: Contribution 1 mirrors prior work
> We agree that individual observations about looped and depth-grown models have appeared separately, and we tried to highlight this in the paper. Our contribution is specifically the **direct, controlled, side-by-side comparison**: the same base architecture, the same data, the same tokenizer, and the same 22-benchmark suite across both model families. The three-axis framing (Fig. 1: unique parameters, inference FLOPs, and training FLOPs) reveals their **complementarity**, a conclusion not derivable from prior work studied in isolation.
>
>
> ### W4: Evaluation results seem noisy / contradict [1]
> **Regarding variability:** We have run additional seeds for the 360M models (see response to reviewer Crin), confirming the robustness of the key trends.
>
> **On Figure 1 (iso-inference, middle panel):** The variation in Figure 1 (middle) is not noise but reflects the actual trade-off structure. Comparing the **iso-inference** plot (middle) with the **iso-parameter** plot (left) makes clear that looped models need to retain a **sufficient number of unique parameters** to match or exceed the baseline on reasoning primitives.
>
> **On the apparent contradiction with [1]:** This is because reasoning primitives consist of different tasks in [1] vs our paper; using the [1]-style task average (code/math d0/d1), the discrepancy completely disappears: at 1.7B, Loop (4x6) exceeds the baseline (41.35 vs. 38.40). For a detailed task performance breakdown, please see: https://ibb.co/M51y9BDs.
>
> **On the architecture choice:** The 1.7B model (24 layers) and 360M model (32 layers) follow the **SmolLM-v1 configuration**. In general, scaling law investigations show low sensitivity of loss to depth-width aspect ratio within reasonable ranges [4], and we compare directly to a tuned baseline setup (SmolLM).
>
>
> ### W5: Figure 2 does not clearly support unified mechanistic signatures
> We agree with the stated observations and note that we argue looped and grown models show **more similar behavior to each other than either does to the baseline**, not that they are identical. From this perspective, the three panels tell a consistent story:
>
> - **Panel A (depth score):** Both looped models and LIDAS score substantially higher than the baseline.
> - **Panel B (top-5 overlap, GSM8K):** The **unique encoder-decoder looped model** Loop(4-4×4-4) tracks LIDAS more closely than Loop(4×6) at later layers. This is consistent with our finding at the end of Section 4.1 (Fig. 5) that the encoder-decoder design recovers depth-grown-like structural robustness.
> - **Panel C (early-exit, Variable Assignment Math):** All non-baseline models continue improving predictions through later layers, whereas the baseline plateaus earlier.
> We will clarify this in the caption of Fig. 2 to make this interpretation more explicit.
>
> ### Minor: Params/FLOPs column in Table 1
> We thank the reviewer for the suggestion.
>
> ### Q1: Inference-time looping on different model families
> Although we have not yet evaluated a different architecture family, we confirmed the same qualitative trend at a second scale within the same family: on SmolLM 360M, depth-grown models benefit substantially more from inference-time looping than the baseline. See: https://ibb.co/Xrr80c00.

---

> > ### Author Rebuttal · Reviewer_wjZC · 2026-04-03
> >
> > Thank you for your rebuttal.
> >
> > Although your work has several interesting findings I still believe that it is premature for publication. As other reviewers stated as well, the current form of the paper is overwhelming - and requires a significant edit which will reduce the amount of experiments and provide a clear and concise manuscript.
> >
> > I will keep my score.

---

> > > ### Author Response · Authors · 2026-04-03
> > >
> > > We thank the reviewer for acknowledging that our work contains "several interesting findings." We note that the concern about a dense scope was also raised by reviewer Crin who recommends strong accept and explicitly states: *"While it is not a flaw, the paper contains a large amount of information"*, and describes the paper as having *"well organized, clear explanation and convincing conclusion."* The reject-leaning assessment therefore stands in contrast not only to the other reviewers' scores but also to their explicit characterization of the same property.
> > >
> > > In our initial response, we addressed each stated weakness directly: we provided an experiment-by-experiment motivation (W1/Q2), clarified the novelty of the side-by-side comparison (W2/W3), provided multi-seed results confirming trend robustness (W4), resolved the apparent contradiction with [1] (W4), gave a detailed panel-by-panel reading of Figure 2 (W5), and confirmed the inference-time looping result at a second scale (Q1). The post-rebuttal response does not engage with any of these points.
> > >
> > > We would be happy to implement actionable feedback specifying what should be changed.

---

### Official Review · Reviewer_rAxJ · 2026-03-13

**Soundness:** 4
**Presentation:** 3
**Significance:** 3
**Originality:** 3
**Overall Recommendation:** 4
**Confidence:** 2

**Summary:**

This paper studies two architectural approaches that have recently been used to improve the reasoning ability of language models: looped transformers and depth-grown transformers. Looped transformers increase the effective computation depth by repeatedly applying the same set of layers during inference, while depth-grown transformers gradually expand model depth during training by duplicating intermediate layers. The central question of the paper is whether these two seemingly different approaches rely on a common underlying computation mechanism.

To investigate this question, the authors first conduct systematic comparisons between standard, looped, and depth-grown models across a variety of reasoning-related tasks, analyzing the trade-offs among parameter count, training compute, and inference compute. The paper further performs several mechanistic analyses, including depth utilization analysis, residual stream interventions, and sublayer contribution analysis, to study how models use different layers during computation. The results suggest that both looped and depth-grown models exhibit similar depth utilization patterns and periodic internal computation structures. Based on these observations, the authors argue that both approaches induce a form of iterative computation within the model.

In addition, the paper shows that applying inference-time looping to depth-grown models can further improve reasoning performance, suggesting that the two approaches can be combined. Based on these findings, the authors propose a practical strategy of “grow first, loop later”, where models are trained using depth growing and then augmented with looping during inference to further enhance reasoning performance while maintaining training efficiency.

**Compliance With Llm Reviewing Policy:**

Affirmed.

**Key Questions For Authors:**

see Weaknesses

**Limitations:**

Yes

**Strengths And Weaknesses:**

Strengths
	1.	The paper studies an interesting and meaningful question. While both looping and depth growing have been used to improve reasoning performance in language models, their relationship has not been systematically examined. Providing a unified perspective on these approaches is a useful contribution.
	2.	The experimental evaluation is relatively comprehensive. The paper not only compares the performance of different model architectures on reasoning tasks but also analyzes the trade-offs among parameter size, training compute, and inference compute, offering a broader view of efficiency and performance.
	3.	The paper includes several forms of mechanistic analysis. Through depth utilization analysis, residual stream interventions, and sublayer contribution studies, the authors attempt to explain why looped and depth-grown models exhibit similar improvements in reasoning ability. These analyses help provide insight into the potential mechanisms behind these architectural designs.
	4.	The proposed practical strategy of “grow first, loop later” provides an interesting empirical guideline that may inspire future work on improving reasoning efficiency in language models.

Weaknesses
	1.	The proposed “unified mechanism” is mainly supported by empirical observations and experimental analyses. While the experiments suggest similarities in model behavior, the theoretical explanation for why both approaches lead to iterative computation remains somewhat limited.
	2.	The experiments are mostly conducted on reasoning-heavy or relatively structured tasks. Evaluating the approach on a broader range of tasks, such as more open-ended reasoning or other language understanding tasks, could further strengthen the generality of the conclusions.
	3.	The experiments are conducted on moderately sized models. Additional validation on larger-scale models would help strengthen the empirical evidence for the proposed conclusions.

---

> ### Author Rebuttal · Authors · 2026-03-31
>
> We thank the reviewer for acknowledging the strengths of our work and recognizing its soundness and potential. We address each of your concerns below and look forward to further discussion upon additional feedback.
>
> >The proposed “unified mechanism” is mainly supported by empirical observations and experimental analyses. While the experiments suggest similarities in model behavior, the theoretical explanation for why both approaches lead to iterative computation remains somewhat limited.
>
> We thank the reviewer for this comment. We would like to note that for looped models, iterative computation follows by definition, repeatedly applying the same set of layers is iterative computation. The reviewer's concern, therefore, reduces to providing a theoretical justification for why depth-grown models also exhibit a form of iterative computation.
>
> The key insight is that at the moment of layer duplication, a depth-grown model is structurally identical to a looped model, i.e., the duplicated middle blocks share exactly the same weights (block similarity = 1). Subsequent training with untied parameters allows the blocks to diverge, but this divergence is empirically limited: block-level weight similarity remains high after training, as shown in [1] (page 4: Fig 2.c) and [2] (page 10: Fig 7.a). Depth growing can therefore be understood as a relaxed form of looping, where the iterative structure is induced through initialization rather than enforced through hard weight tying. We allude to this in our introduction, but we are happy to make this explanatory argument more explicit in the final version of the paper.
>
> > The experiments are mostly conducted on reasoning-heavy or relatively structured tasks. Evaluating the approach on a broader range of tasks, such as more open-ended reasoning or other language understanding tasks, could further strengthen the generality of the conclusions
>
> We thank the reviewer for this suggestion. We performed additional evaluations to address this concern. Specifically, we evaluate on the language understanding and commonsense reasoning benchmark suite from FineWeb (Penedo et al., 2024) [5], which includes HellaSwag, WinoGrande, PIQA, SIQA, OpenBookQA, ARC-Easy, ARC-Challenge, CommonsenseQA, and MMLU. Notably, MMLU alone covers 57 diverse subjects ranging from abstract algebra and formal logic to marketing, nutrition, and world religions, providing a broad test of general language understanding well beyond reasoning-heavy tasks.
> The results (all 1.7B models) confirm that depth-grown models match or slightly exceed the baseline on these broader tasks (LIDAS achieves the highest average at 51.5% vs. 51.4% for the baseline, with MMLU scores of 37.1% vs. 37.8%), while looped models with sufficient unique parameters, e.g., Loop(4-4×4-4), also remain competitive (49.2% average, 36.4% MMLU). This is consistent with our main findings. You can find the full evaluation table here: https://ibb.co/M5n3XBxp
> We would also like to note that our original evaluation suite already spans 22 benchmarks across diverse categories, including closed-book Q&A, open-book Q&A, text completion/language modeling, math word problems, and reasoning primitives, which are not exclusively reasoning-focused (Section 3.2: benchmarks).
>
> > The experiments are conducted on moderately sized models. Additional validation on larger-scale models would help strengthen the empirical evidence for the proposed conclusions.
>
> We acknowledge that validating our findings on larger models would further strengthen the empirical evidence. However, training multiple model variants (standard, looped, depth-grown) at scales significantly beyond 1.7B, each requiring full pre-training runs, is far beyond what is feasible for academic research groups, as the computational costs quickly become prohibitive outside of frontier industry labs with dedicated compute clusters.
> That said, our experiments at 360M and 1.7B already provide meaningful evidence, as we observe consistent trends across both scales. Furthermore, several of the individual components of our work have been independently validated at larger scales: [1] analyze depth-grown models up to 8B, [3] train looped language models (Ouro) at 1.4B and 2.6B parameters on 7.7T tokens, matching the performance of up to 12B standard LLMs, and [4] scale a recurrent-depth looped model (Huginn) to 3.5B parameters and 800B tokens, demonstrating strong reasoning gains through test-time compute scaling.
>
> [1] On the Inductive Bias of Stacking Towards Improving Reasoning; Sanushi et. al, NeurIPS 2024
>
> [2] Do Depth-Grown Models Overcome the Curse of Depth? An In-Depth Analysis; Kapl et. al, Arxiv 2025
>
> [3] Scaling Latent Reasoning via Looped Language Models; Zhu et al., arXiv 2025
>
> [4] Scaling up Test-Time Compute with Latent Reasoning: A Recurrent Depth Approach; Geiping et al., NeurIPS 2025
>
> [5] The FineWeb Datasets: Decanting the Web for the Finest Text Data at Scale; Penedo et al., NeurIPS 2024

---

> > ### Author Rebuttal · Reviewer_rAxJ · 2026-04-04
> >
> > Sorry, due to a lack of relevant background, I cannot provide very detailed suggestions for this work. The authors have addressed all my questions, and I understand the resource constraints in academia. I will keep my score at 4, and hope the work can be further validated and explored in real-world settings.

---

> > > ### Author Response · Authors · 2026-04-04
> > >
> > > We thank the reviewer for taking the time to engage with our responses and for marking all concerns as fully resolved.
> > >
> > > We would like to gently push back on one point. Training multiple 1.7B-parameter models on over 400B tokens is itself a real-world, large-scale setting. Validating these trends at frontier scale (7B+ parameters, trillions of tokens) requires compute resources that are simply not available to academic research groups, and we do not believe such a requirement is a reasonable bar for an academic submission or for the rebuttal period. Moreover, as we noted in our response, the individual components of our work have already been independently validated at larger scales in recent work [1,3,4], and the trends we observe at 360M and 1.7B are consistent.
> > >
> > > Given that all concerns have been fully resolved, we kindly ask the reviewer to take this into account in their final assessment.

---

### Official Review · Reviewer_oLv6 · 2026-03-14

**Soundness:** 4
**Presentation:** 4
**Significance:** 4
**Originality:** 3
**Overall Recommendation:** 5
**Confidence:** 4

**Summary:**

This paper makes the case that two popular ideas for improving language model reasoning, looping the same layers multiple times and making models deeper by copying middle layers during training, are really closely related. The authors argue that both approaches give models a similar kind of step-by-step internal processing, which shows up in the way later layers become more important and certain computation patterns repeat across the network. In experiments, they find that looped models can improve reasoning without increasing parameter count, while depth-grown models can reach similar or better reasoning performance with less training compute. They also show that the two methods work well together, since models trained with depth growth can still benefit from looping at inference time, sometimes leading to large gains on reasoning tasks. Overall, the paper presents a simple idea: these methods are not separate tricks, but two practical ways of encouraging the same underlying iterative reasoning process.

**Compliance With Llm Reviewing Policy:**

Affirmed.

**Key Questions For Authors:**

Do the authors plan to release the codebase for this work?

**Strengths And Weaknesses:**

1.  The paper is clear about what it is trying to establish, and the evidence is broader than a simple benchmark comparison. The core claim is that looping and depth growth share a common iterative-computation mechanism, and the paper supports that with depth-utilization diagnostics, residual-stream interventions, and layer-swapping analyses, not just end-task accuracy. That is consistent with the stated contribution that the two methods exhibit “similar depth-wise computational signatures,” including increased reliance on later layers and block-aligned periodic structure.

2.  The empirical comparison is set up along the right axes. Rather than presenting one method as uniformly better, the paper separates unique parameter count, inference FLOPs, and training FLOPs, and the conclusions are correspondingly more precise. In the authors’ summary, looped models improve reasoning under fixed parameter budgets and can be competitive at fixed inference budgets, while depth-grown models reach similar or better reasoning with less training compute.

3.  The “grow first, loop later” result is one of the more convincing parts of the paper. The authors explicitly show that a depth-grown model can be looped at inference time by repeating a middle block, yielding gains on some reasoning-primitives tasks despite not being trained with weight tying. That makes the unification claim stronger, because it is not just a similarity argument but a compositional result.


Overall, I found this to be an excellent paper and did not identify any significant weaknesses.

---

> ### Author Rebuttal · Authors · 2026-03-31
>
> We sincerely thank the reviewer for taking the time to carefully evaluate our work and for acknowledging both its soundness and significance. We are glad that the reviewer found the evidence supporting our unification claim convincing, in particular the depth-utilization diagnostics, residual-stream interventions, and layer-swapping analyses, and that the “grow first, loop later” compositional result strengthened the argument beyond a simple similarity observation.
>
> Regarding the code release question, our implementation is based on the nanotron framework. We will publicly release the full codebase, training configurations, and model checkpoints to support reproducibility.

---

> > ### Author Rebuttal · Reviewer_oLv6 · 2026-04-06
> >
> > Thank you for the thoughtful clarification and for addressing the code release question. I appreciate the additional details on the empirical evidence for the unification claim, as well as your commitment to releasing the codebase, training configurations, and checkpoints for reproducibility.
> >
> > That said, I will maintain my current score.

---

### Decision · Program_Chairs · 2026-04-30

**Decision:**

Accept (regular)

**Comment:**

This submission studies the relationship between looped transformers and depth-grown transformers, and argues that they can be understood through a unified lens of iterative computation. In addition to a controlled empirical comparison, the paper provides mechanistic analyses and shows that depth-grown models can further benefit from inference-time looping, suggesting the two approaches are complementary rather than competing.

The review set is mixed, with substantial disagreement in scores, but the balance of the discussion is positive. Three reviewers support acceptance, including one strong accept and one clear accept. These reviewers consistently highlight the paper’s technical soundness, the breadth of the experimental study, the value of the controlled side-by-side comparison, and especially the “grow first, loop later” result as a compelling finding. They also find the mechanistic analyses informative and the overall framing meaningful.

The main concerns raised in review were: (1) limited theoretical grounding for the proposed unification, (2) questions about breadth and robustness of evaluation, including variance estimates and broader-task validation, and (3) concern from one reviewer that the paper is overly dense and insufficiently coherent in its current form. In rebuttal, the authors addressed the first two categories substantively. They clarified the intended interpretation of the unified mechanism, arguing that depth growth can be viewed as a relaxed form of looping induced through initialization, and they supported this explanation with references to structural similarity and observed post-training block similarity. They also added broader evaluation on language-understanding and commonsense benchmarks, and provided multi-seed retraining results for the 360M setting that support the robustness of the main empirical trends. Two reviewers explicitly marked their concerns as fully resolved after rebuttal, while maintaining their original positive scores.

The remaining negative review continues to object primarily on presentation and coherence grounds, arguing that the manuscript attempts too many experiments and would benefit from significant pruning and restructuring. The paper is ambitious and somewhat dense, and the final version would benefit from clearer signposting of how the sections fit together and from tightening the narrative. However, I do not view this as sufficient reason to reject the work. First, the concern is not shared by the majority of reviewers; in fact, another reviewer who also noted the density explicitly stated that it is not a flaw and still rated the paper strongly. Second, the negative post-rebuttal discussion did not substantially engage with the authors’ specific clarifications regarding novelty, robustness, or interpretation of the key figures. On balance, I view the remaining concern as one of scope and exposition rather than a fundamental issue of soundness or contribution.

Overall, I find the paper to make a solid contribution. The core question is interesting, the controlled comparison is useful, the mechanistic perspective adds value beyond benchmark reporting, and the compositional result showing that a depth-grown model can benefit from inference-time looping is novel and likely to be of interest to the community. While the manuscript would benefit from a tighter final presentation, the technical contribution appears sound enough and sufficiently interesting to merit acceptance.